# Disruption of myelin structure and oligodendrocyte maturation in a macaque model of congenital Zika infection

Jennifer Tisoncik-Go [1,2,12] ✉, Caleb Stokes [1,3,12] ✉, Leanne S. Whitmore [1], Daniel J. Newhouse[1], Kathleen Voss [1], Andrew Gustin [1], Cheng-Jung Sung [1], Elise Smith[1], Jennifer Stencel-Baerenwald[1], Edward Parker[4], Jessica M. Snyder[5], Dennis W. Shaw[6], Lakshmi Rajagopal [3,7,8], Raj P. Kapur [9,10], Kristina M. Adams Waldorf [7,11] & Michael Gale Jr [1,2,7] ✉

Zika virus (ZikV) infection during pregnancy can cause congenital Zika syndrome (CZS) and neurodevelopmental delay in infants, of which the pathogenesis remains poorly understood. We utilize an established female pigtail macaque maternal-to-fetal ZikV infection/exposure model to study fetal brain pathophysiology of CZS manifesting from ZikV exposure in utero. We find prenatal ZikV exposure leads to profound disruption of fetal myelin, with extensive downregulation in gene expression for key components of oligodendrocyte maturation and myelin production. Immunohistochemical analyses reveal marked decreases in myelin basic protein intensity and myelinated fiber density in ZikV-exposed animals. At the ultrastructural level, the myelin sheath in ZikV-exposed animals shows multi-focal decompaction, occurring concomitant with dysregulation of oligodendrocyte gene expression and maturation. These findings define fetal neuropathological profiles of ZikV-linked brain injury underlying CZS resulting from ZikV exposure in utero. Because myelin is critical for cortical development, ZikV-related perturbations in oligodendrocyte function may have long-term consequences on childhood neurodevelopment, even in the absence of overt microcephaly.

Maternal infection during pregnancy can have severe consequences on fetal development and survival. Zika virus (ZikV) is an emerging flavivirus that can be vertically transmitted to the fetus from an infected pregnant mother, leading to congenital Zika syndrome (CZS), which encompasses a range of fetal malformations including hearing loss, ocular manifestations, intrauterine growth restriction, and microcephaly[1–4], as well as miscarriage[5–7]. CZS persists post-partum and imposes major complications to childhood development, now manifested across ZikV-endemic regions[8,9]. The mechanism of microcephaly in CZS is thought to be related to ZikV infection and death of

[1]Center for Innate Immunity and Immune Disease, Department of Immunology, University of Washington, Seattle, WA, USA. [2]Washington National Primate Research Center, University of Washington, Seattle, WA, USA. [3]Department of Pediatrics, University of Washington, Seattle, WA, USA. [4]Department of Ophthalmology, NEI Core for Vision Research, University of Washington, Seattle, WA, USA. [5]Department of Comparative Medicine, University of Washington, Seattle, WA, USA. [6]Department of Radiology, University of Washington, Seattle, WA, USA. [7]Department of Global Health, University of Washington, Seattle, WA, USA. [8]Center for Global Infectious Disease Research, Seattle Children's Research Institute, Seattle, WA, USA. [9]Department of Pathology, University of Washington, Seattle, WA, USA. [10]Department of Pathology, Seattle Children's Hospital, Seattle, WA, USA. [11]Department of Obstetrics & Gynecology, University of Washington, Seattle, WA, USA. [12]These authors contributed equally: Jennifer Tisoncik-Go, Caleb Stokes. ✉e-mail: tisoncik@uw.edu; caleb.stokes@seattlechildrens.org; mgale@uw.edu

neural progenitor cells leading to decreased neurogenesis[10–12]. However, the pathogenesis of neurodevelopmental delay in CZS, particularly in those displaying normal head circumference, termed normocephalic, is poorly understood. Important questions remain in understanding the impact of ZikV infection on prenatal development, and perhaps chief among these is the question of how ZikV causes neurologic injury in CZS, including among normocephalic outcomes.

Neuronal remodeling and myelination are major processes that account for central nervous system (CNS) growth and maturation[13]. Myelin, an extension of the lipid membrane of oligodendrocytes, wraps around axons and plays a critical role in neuronal function by insulating and facilitating efficient transmission of electrical signals along the axon[14]. In humans, myelination initiates as early as the fifth fetal month within the caudal brain stem and progresses rostrally to the forebrain, with rapid additional development within the first two years of postnatal life[15–17]. The formation of myelin by oligodendrocytes is necessary for the development of complex neurologic circuits that underlie movement, sensory processing, cognition, and memory[18–22]. However, in fetal development, the myelinated axons in the deep cortex are uniquely susceptible to injury by hypoxia and inflammation[23].

Nonhuman primate (NHP) models of ZikV infection in pregnancy recapitulate aspects of vertical transmission, fetal neuropathology, fetal demise, and miscarriage observed in humans[24–29]. We have established an NHP model of congenital ZikV infection in pregnancy wherein maternal to fetal virus transmission can result in fetal neuropathology with microcephaly[30] or without microcephaly[31], reflecting human CZS.

Here, we employ a systems biology approach to characterize CZS in an NHP model, defining fetal demyelinating disease following maternal-to-fetal ZikV transmission in mid-to-late gestation in the context of otherwise normocephalic fetal development. Spatial transcriptomic, bulk mRNA sequencing (RNAseq), magnetic resonance imaging (MRI), histopathologic, and virologic analyses of fetal brain tissue reveal that ZikV-exposed fetuses have extensive changes in white matter histology, gene expression, and specific protein levels occurring independent of microcephaly and are sustained after ZikV RNA is cleared from the tissue. These alterations include genes that span all maturational stages of oligodendrocyte development and reveal specific tissue disorganization with altered oligodendrocyte morphology within brain lesions following fetal exposure to ZikV. The structure of myelin in ZikV-exposed fetuses is perturbed and, in the most severely affected animals, there is evidence of oligodendrocyte injury and axonal dysfunction. These findings argue that oligodendrocyte alteration leading to dysregulation of myelination and myelin wrap maintenance are features of CZS. Since altered myelination in CZS can occur in the absence of microcephaly, our findings implicate oligodendrocyte dysregulation and myelin disruption as an underlying feature of CZS that could impact pre- and post-natal neurologic development in children with CZS.

## Results

In an established NHP model of transplacental Zika virus transmission, we investigated neuropathological changes in fetal white matter after maternal ZikV infection during pregnancy using spatial and bulk tissue transcriptomics, immunohistochemistry (IHC), electron microscopy (EM) and magnetic resonance imaging (MRI) analyses. We conducted a cohort study of 6 maternal ZikV challenge animals who received ZikV subcutaneously at times ranging from 60–121 gestation days (GD), and 6 control animals who received saline (at 59–138 GD) instead of virus challenge (Fig. S1a and Table S1). Animals ZIKA 1 and ZIKA 2 were challenged with ZikV/FSS13025/Cambodia[30], while animals ZikV 3–6 were challenged with ZikV/Brazil/Fortaleza/2015[31] (Table S1). Each fetus was delivered by Cesarean section at gestational ages ranging from 141–159 days, corresponding to the late third trimester (Table S2).

Within the ZikV challenge cohort, transient viremia was demonstrated across 5/6 ZikV-challenged dams at 2 days post-infection (DPI), with ZikV RNA detected in the fetal brain at necropsy of 3/6 ZikV cohort animals (Fig. S1e)[31]. While there was a trend toward smaller brain volume in ZikV-exposed animals, none reached the threshold of >2 s.d. smaller than controls to be considered microcephalic (Table S2).

To define fetal brain transcriptome changes following maternal infection with ZikV, we used spatial transcriptional profiling to identify gene expression patterns from discrete regions of interest (ROIs) in the developing parietal cortex. We chose ROIs representing functionally distinct compartments as follows: deep grey matter (DGM, containing cortical Layer V pyramidal neuron cell bodies), superficial white matter (SWM, containing proximal axons in cortical Layer VI), and deep white matter (DWM, containing myelinated axons of projecting neurons deep to the cortex) (Fig. 1a). ROI-specific gene expression patterns matched those predicted by the predominant cell types in each region (Fig. 1b). With regional signatures identified in healthy controls, we next assessed the impact of ZikV exposure on gene expression (Fig. 1c). Our analysis indicated that the largest magnitude of ZikV-related transcriptional changes occurred in the DWM (Fig. 1d, e, Fig. S2a, b, Tables S3 and S4). The DWM of ZikV-exposed animals compared to control had markedly reduced expression (downregulation) of oligodendrocyte genes fundamental to the formation and maintenance of myelin sheaths in the central nervous system, including *MBP, MOBP, PLP1,* and *CNP* (Fig. 1d). In contrast, the DGM of ZikV-exposed fetal brains showed increased expression (upregulation) of genes underlying axon growth (*NCAM1, TUBB, GAP43*; Fig. 1e), and neuronal immaturity (*SOX11, DCX, SATB2*; Fig. S2c) compared to control. The principal component analysis did not identify transcriptional differences between animals according to the detection of ZikV RNA in fetal tissue or ZikV challenge strain (Fig. S2d).

An over-representation analysis of differentially expressed genes (FDR < 0.05) displayed using gene network analysis across DGM and DWM fetal brain regions demonstrated downregulation of genes linked to oligodendrocyte differentiation and function in the DWM of ZikV exposed fetuses; DGM changes included upregulation of neuron projection guidance, cell migration, and neuron development genes (Fig. 1f, S2b, c and Supplementary Data 3). Upstream regulator analysis of DE genes indicated decreased activity of the transcription factor *TCF7L2*, which controls oligodendrocyte development and myelin-related gene expression[32,33]; importantly, this decrease was predicted in all three brain regions for ZikV-exposed fetuses (Fig. S2e and Supplementary Data 4). This finding agreed with our identification of decreased gene expression across all maturational stages of the oligodendrocyte lineage within ZikV-exposed fetal DWM, including *SOX10 and OLIG2*[34] (Fig. S2f). TCF7L2 expression was downregulated in DWM (p = 0.03; FDR = 0.19). In addition, DGM (and DWM, to a lesser extent) had downregulated genes related to synaptic signaling (e.g., *SYN2, SLC17A7,* and *CPLX1*). Upstream regulator analysis also revealed decreased activity of the transcription factor *SOX2* in SWM and DWM of ZikV-exposed fetuses, consistent with our previous report of reduced Sox2+ cells in neurogenic populations in the subventricular zone (SVZ) of ZikV-exposed fetuses[31].

This spatial transcriptomic analysis resolved ZikV-related gene expression changes within specific regions of the parietal cortex. To examine changes of cell populations in the parietal cortex, we performed bulk RNA sequencing (RNAseq) of superficial fetal cortical samples spanning the anterior-posterior axis and used CIBERSORT[35] to deconvolve gene expression profiles to estimate the relative abundances of cell types in the tissue. The bulk RNAseq analysis demonstrated widespread transcriptional changes related to axon guidance and myelination across the fetal brain (Fig. S3a–c and Supplementary Data 5-6)[32–34]. Cellular deconvolution analysis indicated the proportions of cell types in the fetal cortex of ZikV-exposed animals were largely unchanged relative to controls (Fig. S3d). In animals with ZikV

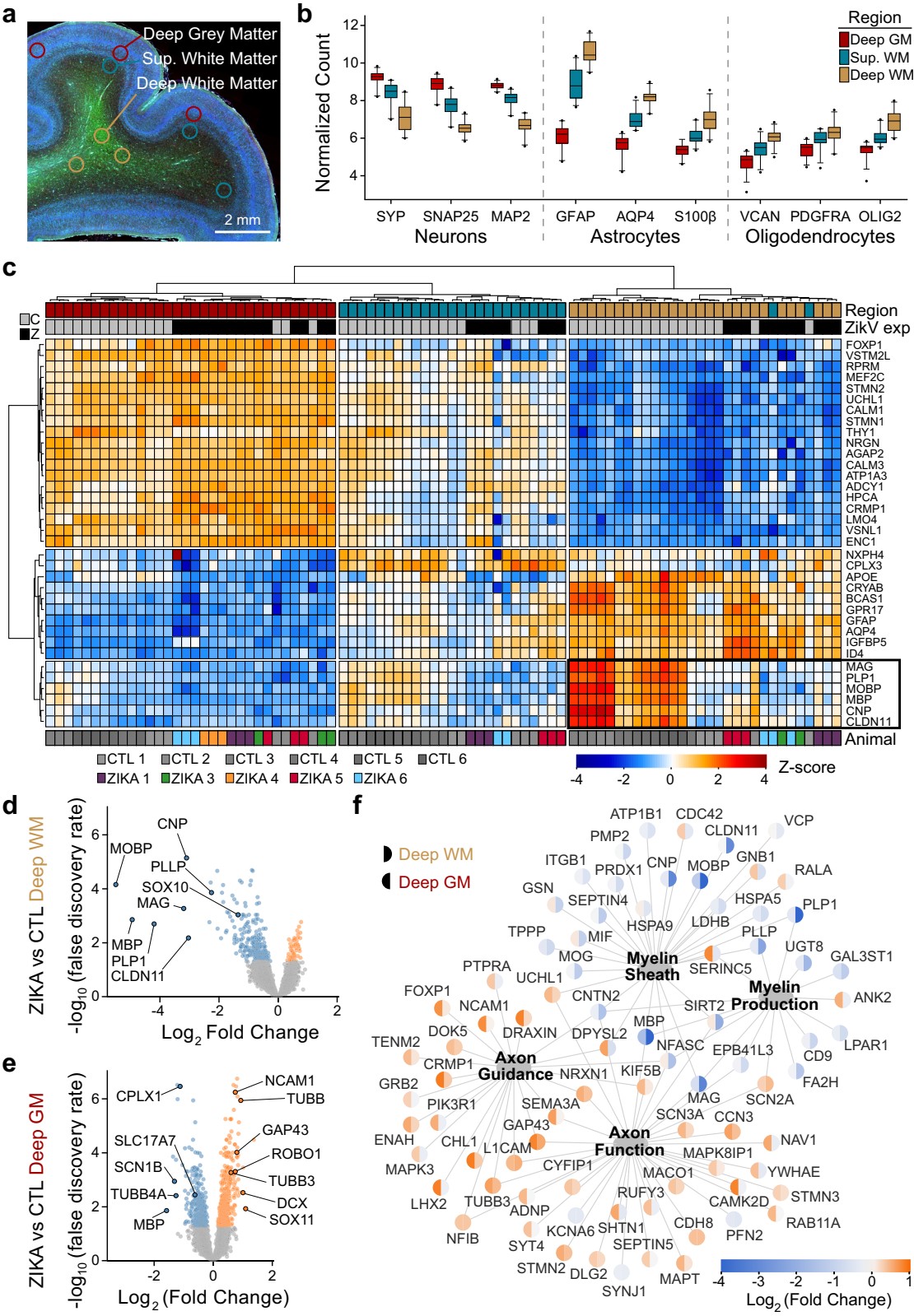

RNA detectable by PCR in the fetal brain, the spatial distribution spanned the parietal and occipital cortex (Fig. S1e, Fig. S3e).

To further define the impact of ZikV exposure on myelin and oligodendrocytes, we analyzed white matter cellular composition by performing immunohistochemistry on the parietal and occipital cortex. The expression of myelin basic protein (MBP), a key structural component of myelin, was significantly diminished in the brains of

ZikV-exposed fetuses compared to controls (Fig. 2a–h, Fig. S4a–d). Qualitatively, this observation corresponded to both a reduction in the MBP staining intensity and the number of MBP+ fibers in most cases (Fig. S4e). Luxol fast blue staining for compact myelin corroborated these findings, with decreased staining of the white matter of ZikV-exposed animals (Fig. S4f). There were no differences between the control and ZikV-exposed fetal brain in the density of cells staining for

**Fig. 1 | Fetal ZikV exposure causes downregulation of myelination genes in deep white matter of nonhuman primate.** Digital spatial profiling (DSP) of tissue from control (*n* = 6) and ZikV (*n* = 6) animals was conducted using the Nanostring GeoMx DSP platform, after immunofluorescence staining to identify regions of brain and cell types. Samples from ZIKA 2 did not meet quality control metrics for downstream analysis. **a** ROIs were selected in triplicate for each brain, representing DGM (red), SWM (teal), and DWM (tan). **b** Box and whisker plots (box = 25th–75th percentile and median, whiskers = 5th–95th percentile) representing normalized counts for selected genes classically expressed by neurons (left), astrocytes (center), and oligodendrocytes (right), according to ROI. Data for graphs are provided in Source Data file. **c** Normalized gene expression (row-specific Z-score) of the top 35 differentially expressed (DE) genes identified in pair-wise comparison of samples across ROI and ZikV exposure. Samples (*x*-axis) and genes (*y*-axis) were clustered by calculating Euclidean distances using Ward.D2. Top row, color coding by ROI, as in (**b**). Second row, color coding by exposure: grey, control; black, ZikV. Bottom row: color coding by animal. Black outline highlights genes enriched in DWM that relate to myelination. **d**, **e** Volcano plots of DE genes comparing ZikV to control animals for ROIs representing (**d**) DWM and (**e**) DGM. Orange, significantly (FDR < 0.05) upregulated in ZikV; blue, significantly downregulated in ZikV; grey, FDR > 0.05 in DE comparison. f Network of 79 DE genes (FDR < 0.05) in either DWM or DGM clustered by gene ontology (GO, large nodes) representing axon function (GO:0030424 and GO:0007411) and myelination (GO:0042552 and GO:0043209). GO terms were selected by applying over-representation analysis (ORA) to DE genes in each cluster (Fig. S2a). Small nodes represent average log-fold change (color) for each gene in DWM (left half) and DGM (right half). Average gestational age (±SD) of ZikV-exposed vs CTL animals in DSP analysis = 150(±9) vs 156(±2) days; *p* = 0.14 by *t* test for DGM; 154(±8) vs 156(±2) days; *p* = 0.46 by *t* test for DWM. Source data are provided as a Source Data file.

Olig2 (Fig. 2i), which labels both oligodendrocyte precursors and myelinating oligodendrocytes[36,37]. There also were no significant differences in the abundance of astrocytic marker, glial fibrillary acidic protein (GFAP), or microglial marker, allograft inflammatory factor 1 (AIF-1/Iba1), in the parietal cortex with respect to ZikV exposure (Fig. S4g-j), or the density of NeuN-positive neurons in any layers of cortex, either in occipital or parietal cortex (Fig. S6). However, we found a local increase in the GFAP intensity and microglia density in the ependymal lining of the posterior lateral ventricle of ZikV-exposed animals, corresponding to a T2-bright primary lesion on MRI (Fig. S5). We also noted a transition zone between ciliated ependymal cells and smooth columnar epithelium, with underlying disruption in the cellular architecture (Fig. S5d-f). This observation is consistent with previous descriptions of increased GFAP-immunoreactive gliosis and microglial activation in the periventricular region within the central nervous system of a pregnant rhesus macaque ZikV infection model[26,30,31]. There were no significant differences in immunohistochemical quantification of MBP, GFAP, or Iba1 when comparing ZikV-exposed animals based on the detection of ZikV RNA in fetal tissue or ZikV strain inoculated (Fig. S1e).

To assess the spatial and temporal extent of the pathophysiological changes in ZikV-exposed animals, we reviewed serial magnetic resonance imaging. We did not observe MRI evidence of intracranial calcifications, cortical malformations, corpus callosum dysgenesis, or vermian hypoplasia in any fetus. In addition to a previously described T2-bright posterior periventricular (primary) lesion in ZikV-exposed animals[31], we noted T2-weighted signal abnormalities in the subcortical white matter that were absent in age-matched controls (compared at average GD123) and appeared to be persistent across multiple imaging time points (Fig. 3a–c, see Fig. S1). These findings were most pronounced in parietal and occipital regions corresponding to the primary sensory and visual areas of cortex. As T2-weighted signal hyperintensity in white matter can reflect diverse pathologic processes including abnormal myelin structure, delayed myelination, or inflammation[38], we performed detailed histological and electron microscopic (EM) analysis. On hematoxylin and eosin (H&E) staining of white matter, we did not observe any evidence of inflammatory infiltrate. While we noted vacuolar changes in the deep white matter of both groups, there was a trend toward more severe vacuolation in the white matter of ZikV-exposed animals than in controls (Fig. 3d, e). In the DGM overlying the site of the primary periventricular lesion, EM revealed variable disruption of the brain parenchyma that was not observed in control animals, while in parietal grey matter there were less severe changes to ultrastructural architecture (Fig. S5g).

We further used EM to analyze the ultrastructural characteristics of axons in the white matter of the parietal cortex in control and ZikV-exposed fetal brain (Fig. 4). In both groups, most large-diameter axons had a compact myelin sheath, with no consistent difference in axon diameter (Fig. S7a–e). However, the myelin in ZikV-exposed animals had numerous focal areas in which the laminar structure was disrupted, with outward bowing of the sheath and widened interlamellar spaces filled with electron-dense material (Fig. 4b). We refer to this finding as myelin decompaction, as it structurally resembles a phenotype that has been described in animal models of axonal injury and in knockdown of myelin structural proteins such as MBP[39–42]. In many areas of decompacted myelin, we observed swelling of the inner lamella of the myelin sheath. We did not find evidence of myelin phagocytosis or increased density of phagocytes or other immune cells within the white matter (Fig. S4h). Intact regions of myelin had apparently normal ultrastructural properties, including number of wraps and wrap thickness, as expected based on gestational age (Fig. 4c, d, Fig. S7f, g). However, there was a significantly higher proportion of axons with myelin decompaction in ZikV-exposed animals as compared to controls (Fig. 4e). We measured the myelin g-ratio, which describes the fraction of the axon diameter composed of myelin, and may be increased in demyelinating conditions[43] or decreased in hypomyelinating conditions[19]. There were no consistent differences in the g-ratio or the slope of the regression line relating the g-ratio to axon diameter across control and ZikV-exposed animals. However, in the most severely affected ZikV-exposed animal (ZIKA3), we found a reduced g-ratio (Fig. 4f, g, Fig. S7d). Moreover, we found swelling of axonal mitochondria in ZIKA3 that is suggestive of axonal stress or injury (Fig. S7h, i). Overall, there were no differences in fetal disease phenotype across ZikV animals following exposure to either ZikV strain used in our studies, showing that myelin perturbation is not strain specific for Asian lineage ZikV.

Together, these data demonstrate oligodendrocyte and myelin perturbation in ZikV-exposed prenatal macaque brain spanning scale from altered gene expression to changes in cellular structure and function, revealing linkage between myelin disease and neuronal dysfunction in CZS that would have profound consequences on childhood neurodevelopment after fetal exposure to ZikV, even in normocephalic infants.

## Discussion

Neurotropic ZikV virus emerged to global importance as an etiologic agent of microcephaly and is now recognized to cause CZS, characterized by extensive motor and cognitive impairment in developing children[44,45]. While microcephaly is a hallmark of severe CZS, additional neuroanatomic abnormalities have been defined using neuroimaging, including reports of delayed myelination[46]. Moreover, normocephalic infants with in utero ZikV exposure may have higher rates of neurodevelopmental delay[47–50]. Here, we challenged pregnant pigtail macaques in mid-gestation with Asian or American lineage ZikV, revealing similar normocephalic CZS phenotype of myelin perturbation across the fetal brain of all ZikV-exposed fetuses. This NHP model reveals several key points: i) white matter injury in CZS is not ZikV strain-specific but may manifest from different viral strains following fetal

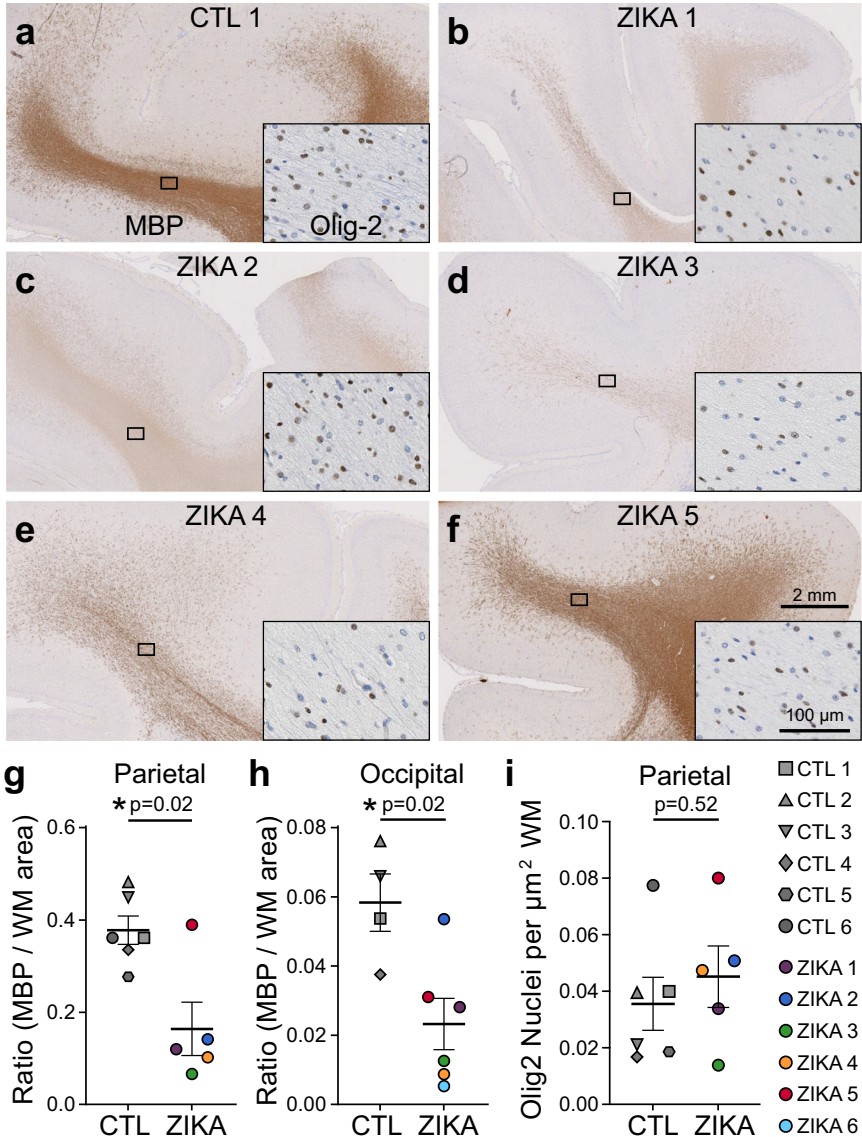

**Fig. 2 | Immunohistochemical analysis demonstrates marked reduction in myelin basic protein (MBP) in ZikV-exposed fetal NHP brains. a–f** MBP (primary image) and Olig2 (inset) immunohistochemical staining of (**a**) control and (**b**–**f**) ZikV animals. The representative images are taken from the dorsal parietal cortex; the black rectangle identifies the approximate location in the DWM tracts represented in the inset. Scale, shown in (**f**), is identical across primary images and across inset images. **g**, **h** Quantification of MBP staining in the DWM from (**g**) superior gyri of parietal cortex and (**h**) inferior gyri of occipital cortex, measured as the ratio of area occupied by chromogen divided by the area of the DWM region quantified (see

Fig. S4a–e). **i** Quantification of the density of Olig2+ nuclei within the DWM. Points in the plots represent individual animals, with bars indicating mean ± SEM. Data for graphs are provided as a Source Data file. Sections included for comparison were matched by rostro-caudal location and stained in parallel. *N* = 6 CTL and 6 ZikV animals; as ZIKA 6 did not have tissue matching the parietal location available, it was only included in the occipital analysis. *P*-values reflect unpaired *t* test with Welch's correction. Average gestational age (±SD) of ZikV-exposed vs CTL animals in IHC analysis=152(±2) vs. 157(±9) days; *p* = 0.23 by *t* test. Source data are provided as a Source Data file.

exposure from maternal ZikV infection[51], and ii) myelin decompaction mirrors injury patterns seen in humans, offering a window into the pathophysiological mechanisms underlying CZS[26,30,31,52]. We found widespread and severe disruption of CNS myelin in fetuses (5 female/1 male) that were normocephalic and had no overt neuroanatomic abnormalities at birth. Our systems biology transcriptomic analysis involved multi-scale systematic characterization of the brain from the ZikV cohort animals, revealing altered gene expression in oligodendrocyte and neuronal development, reduction of myelin proteins, and myelin decompaction. These observations indicate that ZikV exposure can induce a demyelinating disease during prenatal development, which is a feature contributing to CZS.

Due to the widespread white matter injury pattern observed in the fetal brain, we propose that maternal-to-fetal transmission and infection

with ZikV disrupts fetal myelin through a direct or indirect virus-imposed blockade on fetal oligodendrocyte function and maturation (Fig. S8). We were able to directly confirm ZikV infection in 3/6 fetuses, while two of three fetuses in which we did not detect ZikV RNA had MRI findings demonstrating a primary lesion in the posterior periventricular region, the niche of neural progenitor cells (NPCs)[31], arguing that they were infected with ZikV but fetal brain infection was cleared at the time of our analysis. NPCs are a primary cellular target of ZikV in the fetal brain, and NPC infection by ZikV disrupts cortical neuron migration[27]. In a fetal baboon model of congenital Zika infection, Gurung and colleagues found a decrease in oligodendrocyte precursor cells (OPCs) in the cerebellum[53]. Moreover, in a mouse model of ZikV infection, ZikV infects glial progenitor cells, perturbing OPC proliferation and differentiation[54–56]. Similarly, our spatial transcriptional profiling data

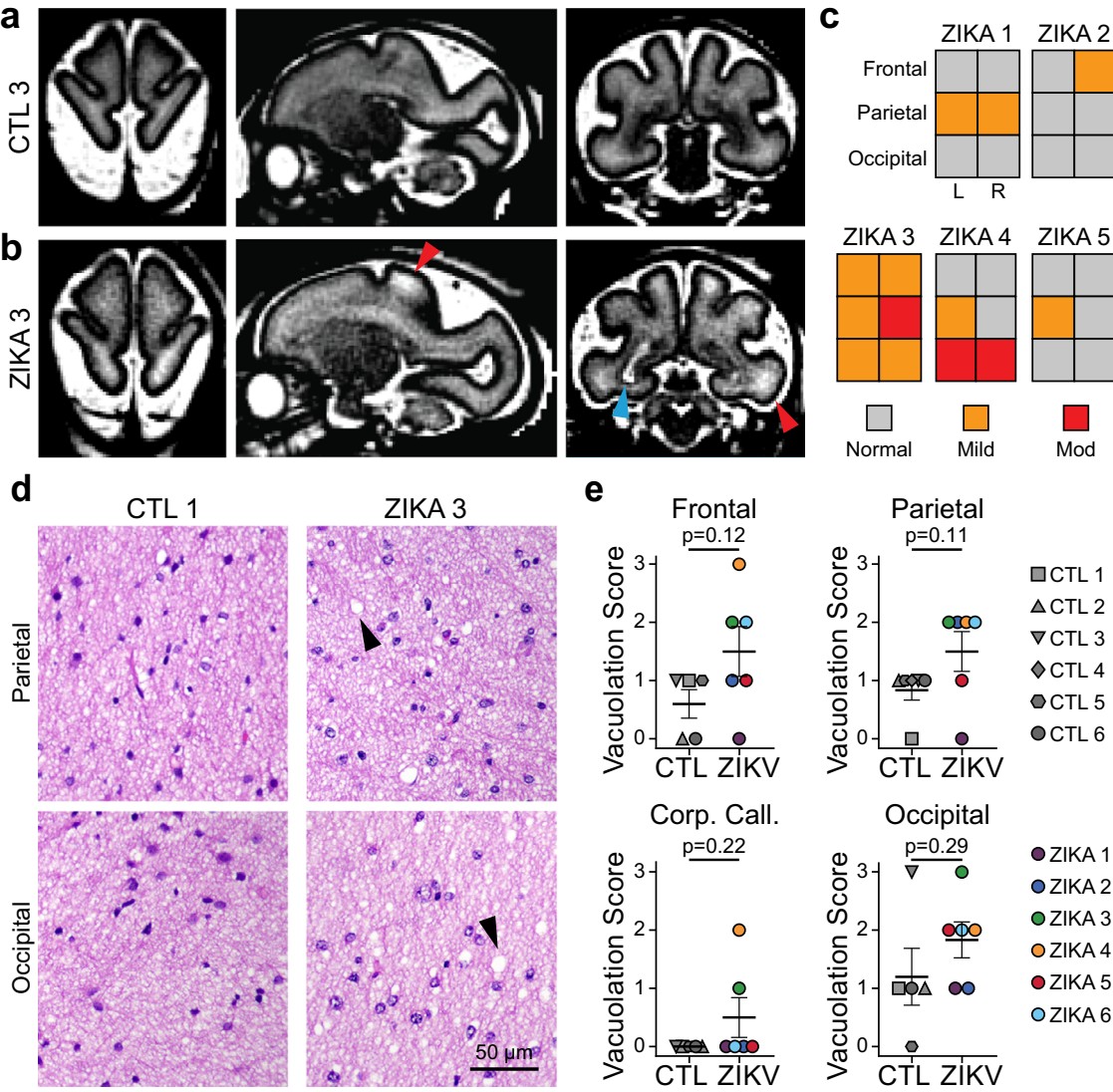

**Fig. 3 | Spatial heterogeneity of pathology in ZikV-exposed fetal NHP brain.**
**a**, **b** Representative T2-weighted MRI images (left, axial; middle, sagittal; right, coronal) for (**a**) control and (**b**) ZikV animals. MRI scans were collected at GD125 for CTL3 and GD120 for ZIKA3 (Fig. S1). Blue arrowhead indicates the primary peri-ventricular lesion described previously[31]. Red arrowheads indicate T2-weighted (T2W) hyperintense foci in subcortical white matter, which can be seen in delayed, abnormal, or inflamed myelin. **c** Ordinal scale of T2W intensity identified in the frontal, parietal and occipital lobes of Zika-exposed animals. Relative scale was established with normal appearance matching control animals and severe signal abnormality matching the T2W signal of the primary lesion. **d** Representative hematoxylin and eosin (H&E) images of white matter at the level of parietal lobe

(top) and occipital lobe (bottom) for control (left) and ZikV animals (right). Arrowheads indicate areas of vacuolization seen in the H&E staining of white matter. **e** plots representing semi-quantitative scoring of white matter vacuolar changes (concordance of two blinded evaluators; n = 6 CTL and 6 ZikV animals) performed on H&E staining of deep white matter from frontal, parietal, and occipital brain regions as well as corpus callosum. Vacuolar changes were scored from 0–3 using an ordinal system; grade 0 indicated normal tissue and grade 3 indicated moderate vacuolation over at least 2/3 of the tissue area. Data for graphs are provided as a Source Data file. Error bars, mean ± SEM. Source data are provided as a Source Data file.

revealed a decrease in the expression of OPC-specific transcription factors *OLIG2* and *SOX10* in the ZikV fetal brain cohort (see Fig. S8). This observation also validates our previous work demonstrating a decrease in Sox2 from the subventricular zone within the NPC niche[31]. Here, fetal exposure to ZikV did not result in changes in the density of Olig2+ cells in white matter. In some conditions of neurologic injury, Olig2+ OPC populations expand and differentiate to oligodendrocytes, facilitating repair and re-myelination of injured axons[57–59]. However, the expansion of OPCs in these instances leads to upregulation of Olig2 and other early markers of the oligodendrocyte lineage. In contrast, we observed downregulation of genes spanning the oligodendrocyte lineage (including *OLIG2* and *SOX10*), suggesting a broader mechanism of oli-godendrocyte dysregulation by ZikV, impairing cell maturation as well as myelin production.

A range of mechanisms have been identified underlying demye-linating diseases, including direct insult on developing oligoden-drocytes, loss of trophic support from axonal degeneration, and immune-mediated attack. Fetal white matter is uniquely susceptible to injury. Hypoxia or infection in utero can cause periventricular leuko-malacia (PVL), in which necrotic death of pre-myelinating oligoden-drocytes is accompanied by astrogliosis and microglial activation[60,61]. Although we identified local disruption of tissue architecture and gliosis at the site of the posterior periventricular ZikV fetal brain lesion, we found extensive changes to myelin at distal sites throughout the brain, without necrosis or microglial activation typically observed in focal PVL. In adults, ZikV infection has been associated with auto-immune attack on myelin, including Guillain-Barré syndrome and acute myelitis, wherein ZikV was cultured from a patient with

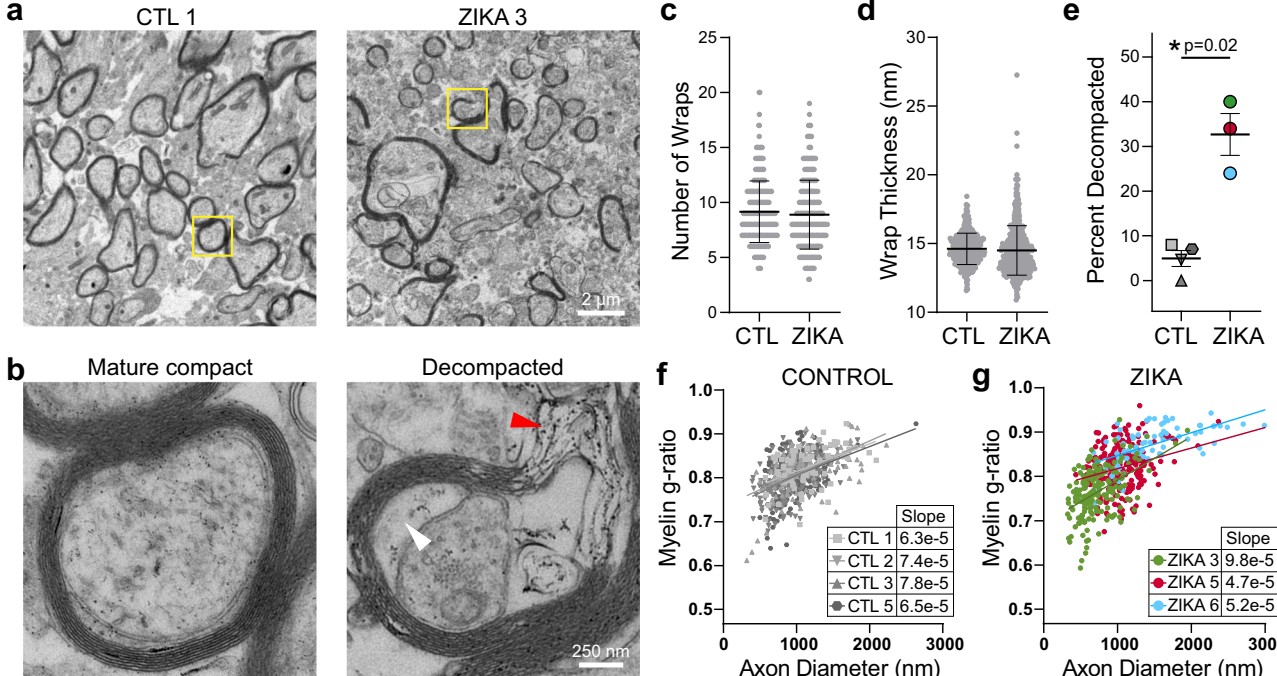

**Fig. 4 | ZikV exposure is associated with myelin sheath decompaction in fetal NHP brain.** EM analysis of control (*n* = 4) and ZikV (*n* = 3) brain sections sampled from DWM in the parietal cortex. **a, b** Representative EM images of myelinated axons from a control (left) and ZikV-exposed animal (right). The area marked by the yellow rectangle is expanded in (**b**), demonstrating mature compact myelin (left) and decompacted myelin (right) with electron-dense material in the interlamellar space (red arrowhead) and swelling of the inter lamella (white arrowhead). **c–e** Quantification of specific myelin features including the number of (**c**) myelin wraps and (**d**) average wrap thickness measured in areas of compact myelin. Individual points represent data for a single axon; error bars represent mean ± SEM across all points within a treatment condition. **e** Percent of axons demonstrating a decompaction phenotype as defined by delamination of all layers of the myelin sheath with outward bowing, affecting at least 25% of the circumference of the axon. Individual points represent the percent of myelinated axons with decompaction from one animal; error bars represent mean ± SEM across all animals within a condition. *P*-value calculated by unpaired t-test with Welch's correction; 621 CTL axons and 488 ZikV axons. **f, g** Plot of myelin g-ratio (outer diameter of axon divided by outer diameter of myelin sheath, measured in areas of compact myelin) versus axon diameter. Individual points represent a single myelinated axon. Linear regression was calculated for all points representing a single animal; slope is indicated in the table. Scale, shown in (**a, b**), is identical across primary images. Source data are provided as a Source Data file.

meningoencephalitis[62–65]. Moreover, a fatal case of encephalitis in a non-pregnant woman infected with ZikV was linked with auto-antibodies against myelin oligodendrocyte glycoprotein (MOG)[66]. However, in our study there was no histopathologic evidence of inflammatory infiltrate and minimal induction of proinflammatory pathways from the transcriptomic data in the fetal brain of the ZikV cohort animals, indicating the demyelinating phenotype is not the result of a chronic T cell-mediated autoimmune inflammatory response against myelin proteins or phagocytic attack of oligodendrocytes. It remains possible, however, that diffusible signals from the inflammatory response at the primary lesion in the brain or from other fetal tissues led to widespread perturbations in oligodendrocyte lineage maturation signaling—a mechanism that has been proposed to explain diffuse white matter injury in PVL[67,68].

Several animal models of CNS injury have described a similar phenotype of myelin decompaction, most notably the optic nerve crush model that is used to study demyelination and axon regeneration[39,69,70]. While the precise mechanisms of myelin decompaction are areas of active investigation, acute knockdown of myelin structural components (e.g., MBP) can lead to a similar phenotype[42], suggesting that active signaling and protein synthesis are necessary to maintain compact myelin. Oligodendrocyte maturation and myelin synthesis are closely coupled to neuronal maturation and function in a bidirectional manner[71]. Therefore, we propose that the disruption of myelin may be related to a loss of trophic or maturation signals derived from local neurons or even astrocytes[41] (Fig. S8). Indeed, our spatial transcriptional data from deep grey matter shows a decrease in the expression of genes for synaptic function (*CPLX1, SLC17A*) and an increase in genes associated with immature neurons (*SOX11, DCX, SATB2*). We note that in a mouse model of flavivirus encephalitis recovery, ZikV infection leads to loss of synapses[72]. The gene networks we observed in DGM may therefore represent remodeling of neuronal circuits in response to the loss of synapses. This type of developmental neuroplasticity is a well-described phenomenon in which neurons that experience a loss of functional connectivity undergo axonal outgrowth in order to find new synaptic partners[73,74], and could be a widespread response to focal ZikV infection in the fetal brain affected by CZS[72].

This study has possible limitations that should be considered in the interpretation of our findings. First, the pathophysiology of congenital ZikV infection in NHP models may not fully replicate that in human CZS. In the current study, ZikV-exposed fetuses did not develop gross microcephaly (>2 s.d. below age-corrected head circumference)[31], although they did have smaller brain volumes than controls (Table S2), which has been reported in several other NHP models of ZikV infection in pregnancy[24,26,27]. Necropsy prior to natural birth in NHP models may obscure the development of secondary microcephaly, whereas microcephaly in human infants with CZS can progress after birth[75]. While some NHP models of congenital ZikV demonstrate subtle or no neuropathologic changes[24,76], most demonstrate gross histopathology that closely mirrors human CZS[77]. Second, we were unable to verify fetal infection by PCR in 3/6 ZikV-exposed fetuses, although we confirmed maternal infection in all cases. These observations mirror human data in which ZikV RNA is detectable in less than half of CZS cases at birth[78]. We have proposed a brain-intrinsic mechanism for myelin perturbation, but these experiments do not definitively rule out the possibility of extra-fetal mechanisms including

maternal inflammatory cytokines or placental disruption. Although chronic placental inflammation has been associated with white matter injury in premature infants[79], the only placental pathology observed in ZikV-exposed animals was mild deciduitis, which was also present in some control animals[31]. Third, we present a cohort of 6 ZikV-exposed and 6 control fetuses in which only one animal from each experimental condition was male. This may have precluded our ability to detect subtle neuropathology and/or sex-related changes. Finally, we combined analysis of animals infected with two closely related isolates of ZikV and across a range of gestational ages from first and second trimester. We did not detect strain-related differences in sub-group analysis. Although the study may not have been sufficiently powered to detect small differences, the recapitulation of myelin decompaction in both strains and across time points argues for a conserved pathophysiologic mechanism and suggests that white matter injury may be common in human CZS.

The risk of recurrent ZikV outbreaks in endemic regions due to waning population immunity or new epidemics due to ZikV introduction into naïve populations remains a lingering threat, with a major impact resulting from maternal infections during pregnancy[80,81]. Emerging infectious diseases have the greatest impact on immunologically naïve populations and at risk individuals, such as pregnant women and their unborn, as further demonstrated by the recent SARS-CoV2 pandemic[82,83]. Understanding how ZikV impacts cellular processes during prenatal development is necessary to develop therapeutic strategies for preventing CZS and mitigating ZikV infection. These findings reinforce the serious nature of ZikV infection, and virus infection during pregnancy in general, and the need for effective vaccines or drugs to prevent congenital infections. Our study adds to these findings by providing additional insight into the pathophysiology of CZS following ZikV fetal exposure, including ultrastructural features of myelin decompaction, oligodendrocyte dysregulation, and changes to neuronal function and signaling, that underlie CZS.

## Methods

### Virus
Working stocks of ZIKV/Brazil/Fortaleza/2015 (GenBank no. KX811222) and ZIKV/FSS13025/Cambodia (GenBank no. MH368551) were obtained by plaque-purifying the viruses and amplifying once in C6/36 *Aedes albopictus* cells. Virus was adsorbed to cells in DMEM supplemented with 1% FBS at 37 °C. After 2-h incubation, the inoculum was removed and virus propagated in complete media supplemented with 5% FBS, 2 mM L-Glutamine, 1 mM Sodium Pyruvate, 100 U/mL of Penicillin, 100 μg/mL of Streptomycin, 20 mM HEPES, and 1X MEM Non-essential Amino Acid Solution for 6 days, with media changed at 3 days post-inoculation. Supernatants at 6 days were then collected and centrifuged at 200*g* at 4 °C for 10 min, and frozen in aliquots at −80 °C. Virus stocks were tittered on Vero cells.

### Study design
The nonhuman primate experiments were carried out in strict accordance with the recommendations in the Guide for the Care and Use of Laboratory Animals of the National Research Council and the Weatherall report, "The use of non-human primates in research". The Institutional Animal Care and Use Committee of the University of Washington approved the study (Protocol Number: 4165-02). There were a total of twelve healthy pregnant pigtail macaques (*Macaca nemestrina; Mn*) (Fig. S1). Sex of the fetus was considered retrospectively as an independent variable, but it could not be randomized due to the study design. ZikV inoculation was administered to resemble the bite of a feeding mosquito. ZIKA1 and ZIKA2, received subcutaneous (s.c.) inoculations of ZIKV/FSS13025/Cambodia at five separate locations on the forearms, each with $10^7$ plaque-forming units (PFU) in their mid-late second trimester of pregnancy, while ZIKA3-6 received similar s.c. inoculations of ZIKV/Fortaleza/Brazil/2015 at five

separate locations on the forearms, each with $10^7$ PFU. Six pregnant control animals, CTL1-6, received s.c. inoculations of media alone at five separate locations on the forearms. Cesarean section was performed at least 10 days before the natural due date (-172 days) to enable fetal and dam necropsy (Table S1). Fetal brains were weighed at birth and sectioned (Table S2).

### Fetal brain sampling scheme
Fetal brains were weighed at birth (Table S2) and bisected along the midline. One hemisphere was sectioned from anterior to posterior, yielding up to ten coronal blocks. Samples from the medial half of four coronal sections (P1-3, and O) were immersed in RNAlater (Invitrogen, Carlsbad, CA) and used for bulk RNA sequencing. RNAlater samples were also collected from the rostral-caudal coronal section from the frontal (F) lobe. A coronal section through the middle of the opposite hemisphere, which sampled parietal cortex superiorly, deep thalamic nuclei medial centrally, and temporal cortex inferiorly, was placed into formalin and subsequently embedded in paraffin for histopathology, spatial transcriptomics, and immune histochemistry (IHC) analyses. Samples collected for EM analysis included cerebral white matter and grey matter from parietal and occipital regions (Fig. S1d).

### ZiKV qRT-PCR assay
Viral RNA was quantified in fetal brain tissue using a ZiKV-specific RT-qPCR assay. Tissue RNAlater samples were weighted and homogenized in buffer RLT (QIAGEN, Valencia, CA) using a bead-beater apparatus (Precellys, Bertin Corp., Rockville, MD). RNA was isolated using the RNeasy kit (QIAGEN, Valencia, CA). cDNA was synthesized from 500 ng tissue RNA where available using the iScript Select cDNA Synthesis Kit (Bio-Rad, Hercules, CA) according to the manufacturer's gene-specific priming protocol. Viral RNA was quantified using the TaqMan Universal PCR Master Mix (Applied Biosystems, Waltham, MA) and an Applied Biosystems 7300 Real-Time PCR System using primer and probe sequences designed against the ZiKV prM gene of ZiKV strain FSS13025 (GenBank number: KU955593.1). Forward primer 5'-CCGCTGCCCAACACAAG; Reverse primer 5'-CCACTAACGTTCTTTTGCAGACAT; Probe 5'-AGCCTACCTTGACAAGCAGTCAGACACTCAA. A standard curve was generated by serially diluting a plasmid containing residues 699-2,382 of the Zika virus genome. The standard curve was used to extrapolate the number of copies of ZiKV RNA per mg of tissue. To adhere to stringent guidelines, Ct (cycle threshold) values > 38 were deemed as not reliably detected and were not reported.

### Digital spatial profiling
Fixed, paraffin-embedded NHP fetal brain sections representing parietal cortex were prepared according to the GeoMx-DSP Slide Preparation User Manual (NanoString, Inc, MAN-10115-04). Unstained 5μm-thick tissue sections from control (n = 6) and ZiKV (n = 6) animals were mounted on Colorfrost microscope slides (Fisher Scientific) and used for GeoMx Digital Spatial Profiling (DSP; NanoString, Inc. v2.5) assay. RBFOX3 (NeuN), GFAP, and Olig2 cellular markers were used to characterize the tissue morphology and select regions of interest (ROIs) for profiling (Table S3). In situ hybridizations were performed with the GeoMx Human Whole Transcriptome Atlas Panel (WTA, 18,676 total targets) according to the manufacturer's instructions. One slide at a time, probes were added to each slide in a hybridization chamber, covered with a coverslip, and incubated at 37 °C overnight. Following incubation, the slides were washed to remove unbound probe and blocked in 200 μl Buffer W and incubated in a humidity chamber. Rabbit polyclonal anti-Olig2 antibody (Millipore Cat # AB9610) was incubated first at 1:100 in Buffer W, followed by Goat anti-rabbit AF647 (ThermoFisher Catalog #A27040) for visualization. The remaining morphology markers were collectively diluted in Buffer W at the following concentrations: 1:50 RBFOX3 (NeuN) (Abcam EPR12763 Catalog #ab190195), 1:400 GFAP (Novus GA5, Catalog # NBP-

33184DL594, and STYO 83 for nuclei visualization for a total volume of 200 μl per slide. Each slide was scanned with a 20X objective and default scan parameters and de-identified to allow blinded data acquisition. For each tissue section, geometric 500 μm diameter circle ROIs were placed in the following regions based on assessment by a Pathologist 1) subcortical (superficial) white matter (SWM; $n = 3$/section), (2) deep WM tracts (DWM; $n = 3$/section), and 3) deep cortical (layer IV–VI) grey matter (DGM; $n = 3$/section). After ROI placement, the GeoMx DSP instrument photocleaves the UV cleavable barcoded linker of the bound RNA probes from each ROI and collects the individual segmented areas into separate wells in the DSP collection plate.

## Library preparation and generation of expression matrices

Total target counts per DSP collection plate for sequencing were calculated from the total samples areas ($\mu m^2$). For sequencing of whole transcriptome analysis (WTA) libraries, the target sequencing depth was 100 counts/$\mu m^2$. Sequencing libraries were generated by polymerase chain reaction (PCR) from the photo-released indexing oligos and ROI-specific Illumina adapter sequences and unique i5 and i7 sample indices were added. Each PCR reaction used 4 μl of indexing oligos, 4 μl of indexing primer mix and 2 μl of NanoString 5X PCR Master Mix. Thermocycling conditions were 37 °C for 30 min, 50 °C for 10 min, 95 °C for 3 min; 18 cycles of 95 °C for 15 s, 65 °C for 1 min, 68 °C for 30 s; and 68 °C 5 min. PCR reactions were pooled and purified twice using AMPure XP beads (Beckman Coulter, A63881) according to manufacturer's protocol. Pooled libraries were sequenced at 2×27 base pairs and with the dual-indexing workflow on an Illumina NovaSeq. Reads were trimmed, merged, and aligned to retrieve probe identity, and the unique molecular identifier of each read was used to remove PCR duplicates converting reads to digital counts for each target within an individual ROI.

## Analysis of spatial RNA sequencing data

Counts from each ROI were quantified using the NanoString GeoMx NGS Pipeline. For the ROI analysis, initial quality control was implemented by first identifying low performance probes by dividing the geometric mean of a single probe count across all samples against the geometric mean of all the probe counts for that gene. All probes >0.1 were kept for analysis, as recommended by NanoString (https://bioconductor.org/packages/devel/workflows/vignettes/GeoMxWorkflows/inst/doc/GeomxTools_RNA-NGS_Analysis.html). To identify samples with high background noise (nospecific probe binding), we first calculated the limit of quantification (LOQ), which is 2 standard deviations above the geometric mean of the negative probes for each sample. The percentage of genes detected above the LOQ value was then calculated and samples removed from the analysis if they fell below a 1% gene detection rate. Additionally, we examined the ratio of the Q3 quartile value against the mean of the geometric mean of the negative probe counts and removed samples with a ratio less than 1, suggesting the signal from the probes in that sample are unreliable. This left 94 ROIs for downstream analysis. Gene counts were normalized using Q3 normalization. Differential expression within each ROI type (DWM, SWM and DGM) was calculated for each gene using a linear mixed effect model with the Geomx Tools R package (doi: 10.18129/B9.bioc.GeomxTools; Supplementary Data 1), using ZikV exposure as the test variable, with random slope and random intercept for animal ID. This method provides an unadjusted p-value for each gene comparison as well as a false-discovery rate (FDR), which is calculated using the Benjamini-Hochberg method. Gene set enrichment analysis was performed using FGSEA[84] or SetRank[85] with Gene Ontology (GO) biological processes on each set of significant DE genes for each region using log fold changes as the ranking metric (Supplementary Data 2). Gene ontology was identified using the C5 ontology gene set GSEA | MSigDB (https://www.gsea-msigdb.org/gsea/msigdb/human/collections.jsp#C5). For gene

network analysis, a subset of significantly enriched pathways (FDR < 0.05 in at least one comparison; Supplementary Data 1) identified from spatial DE analysis of DWM and DGM were visualized in a gene network using Cytoscape v3.9.1 using Omics Visualizer Viz PieChart plug-in[86] (Fig. 1f, Fig. S2d).

## Bulk RNA sequencing

RNA sequencing was performed on 5 individual tissue samples collected for each animal from control ($n = 4$) and ZikV ($n = 5$) groups. For each animal, brain was sampled from the rostral-caudal level of five coronal sections and designated as frontal (F), parietal (P1-3), and occipital (O) (Fig. S1c, d). The parietal designations encompassed midline structures, including thalamus and deep nuclei, as well as temporal lobe. Brain samples were immersed immediately in RNA-Later, stored at 4 °C for 24 h, and subsequently homogenized in QIAzol (QIAGEN). RNA was isolated from QIAzol homogenates following the QIAGEN RNeasy protocol. Ribosomal RNA (rRNA) was depleted from each RNA sample using the Ribo-ZerorRNA Removal Kit (Epicentre), designed for human, mouse and rat samples, but is also effective in reducing rRNA amounts for NHP total RNA samples. Libraries were prepared from 150 ng of rRNA-depleted RNA following the KAPA Stranded RNA-Seq with RiboErase workflow for Total RNA-Seq libraries (KAPA Biosystems). Library quality was evaluated using the Qubit® 3.0 Fluorometer and the Agilent 2100 Bioanalyzer instrument. Constructed libraries were sequenced on a NextSeq 500 Illumina platform, producing 2x75nt stranded paired-end reads. Quality control of the primary sequencing data was performed using FastQC. Ribosomal RNA reads were removed computationally using Bowtie2[87], with an index composed of human, mouse and rat rRNA sequences, resulting in over 30 million reads. Sequence reads were trimmed to 50 bp and then aligned to the pig-tailed macaque (*Macaca nemestrina*) genome (Mnem_1.0) using STAR[88]. Alignment results show >90% mapping of NHP reads to the pig-tailed genome.

## Analysis of bulk RNA sequencing data

Statistical processing and analysis of RNA-seq count data was done with the R statistical computing environment (R Core Team 2019). Gene counts were filtered by a row mean of 3 or greater and then normalized using edgeR to implement TMM normalization[89,90]. Counts were transformed into log-counts for use in a linear model using voom[91]. Principal Component Analysis was performed using factoextra. Differential expression (DE) analysis compared each ZikV (BRZ or FSS) brain sample against its designation-matched CTL sample based on a linear model fit for each gene using Limma[92]. Criteria for all DE analyses were an absolute fold change of 1.5 and an adjusted *P*-value < 0.05 calculated using a Benjamini-Hochberg correction. The average log2 fold changes (LFC) of significantly DE genes for each brain region were averaged between BRZ and FSS to illustrate general expression trends across the two ZikV infections (Fig. S4c). Hierarchical clustering was performed on average LFC for DE genes identified in at least one contrast and over representation analysis (ORA) was performed on each of the clusters using SetRank[85], KEGG, Wiki-Pathways, and Gene Ontology (GO) databases. All gene names were converted to human orthologs and a pathway was considered significantly enriched with an FDR < 0.05 (Supplementary Data 5). CIBERSORTx was used to predict cell type abundances in each brain sample by inputting TMM log2 normalized expression values using the single cell reference data set from Darmanis et al. 2015 (Fig. S4d, Supplementary Data 7)[35,93].

## H&E staining and histopathologic survey

Fixed tissues 4 μm thick from cortex of control ($n = 6$) and ZikV ($n = 6$) animals were sectioned from paraffin blocks and stained with hematoxylin and eosin (H&E) for a histologic assessment. Blinded analysis of white matter vacuolar changes was performed by two Pathologists

from four independent regions 1) frontal corpus callosum (striatum, rostral to thalamus), 2) frontal deep white matter in trigone adjacent to caudate, 3) parietal white matter (lateral to either head or tail of caudate), and occipital (trigone at end of C-shaped ventricle) regions where available. White matter vacuoles were scored in 3 representative fields using a 40x objective, and graded using an ordinal scale of 0–3 with grade 0 indicating normal appearance and grade 3 showing moderate vacuolation involving at least 2/3 or the tissue area.

### Automated Immunohistochemistry staining

Immunohistochemistry (IHC) staining was performed for GFAP, Iba1, MBP, NeuN, and Olig2 (Table S3) utilizing the Leica Bond Rx Automated Immunostainer (Leica Microsystems, Buffalo Grove, IL). Unless otherwise specified all reagents were obtained from Leica Microsystems. Slides were first deparaffinized with Leica Dewax Solution at 72 °C for 30 sec. Antigen retrieval was performed on all slides stained for Iba1 and NeuN with citrate, pH 6, at 100 °C for 10 min and Olig2 stained slides for 20 min. Antigen retrieval was performed on all slides stained for MBP with EDTA, pH 9, at 100 °C for 20 min. Additionally, antigen retrieval for GFAP consisted of proteinase K digestion at 37 °C for 5 min. All subsequent steps were performed at room temperature. Initial blocking consisted of 10% normal goat serum (Jackson ImmunoResearch, Catalog Number 005-000-121) in tris-buffered saline for 20 min. Additional blocking occurred with Leica Bond Peroxide Block for 5 min. Slides were incubated with GFAP (1:500), Iba1 (1:500), or Olig2 (1:500) primary antibodies in Leica Primary Antibody Diluent for 30 min. Next, a secondary antibody, goat anti-rabbit horseradish peroxidase polymerized antibody, was applied for 8 min. Slides incubated with MBP (1:500) primary antibody in Leica Primary Antibody Diluent for 30 min was followed by application of a rabbit anti-rat secondary (Vector Laboratories, Catalog Number AI-4001) for 8 min. Slides incubated with NeuN (1:500) primary antibody in Leica Primary Antibody Diluent for 30 min was followed by application of the Leica Post-Primary linker for 8 min. NeuN-stained tissues were then incubated with a tertiary antibody, goat anti-rabbit horseradish peroxidase polymerized antibody, for 8 min. All antibody complexes were visualized using DAB (3,3'-diaminobenzidine), detection 2X for 10 min. Tissues were counterstained with hematoxylin for 4 min followed by two rinses in deionized water. Slides were removed from the automated stainer and dehydrated through graded alcohol to xylene. Once dehydrated, slides were coverslipped with a synthetic mounting media and imaged.

### Quantitative microscopy and image analyses

Slides were scanned in brightfield with a 20X objective using the NanoZoomer Digital Pathology System (Hamamatsu City, Japan v2.5). The digital images were imported into Visiopharm software (Hoersholm, Denmark v2023.01), and investigators were blinded to animal identity for analysis. Using the Visiopharm Image Analysis module, regions of interests (ROI) were automatically detected around the entire tissue section. For MBP quantitation, ROIs were manually drawn around subcortical deep white matter contained within superior gyri, inferior gyri, or deep projection tracts (Fig. S4a). Slice location along the rostro-caudal axis was identified from reference images of Macaca mulatta available in the Scalable Brain Atlas (https://scalablebrainatlas. incf.org/macaque/DB09)[94–96]. For primary statistical analysis (Fig. 2g, h) sections were only included if they matched rostro-caudal and ROI location across animals, and had been stained in parallel. Using Visiopharm software, the digital images of the MBP- and Iba1-stained slides were converted into grayscale values, using feature band RGB – G with a mean 3 ×3 pixel filter to detect tissue area and feature band HDAB – DAB was used to detect the area of immunostaining. For GFAP, feature band Eosin was used to detect tissue area and HIS-I was used to detect the area of immunostaining. For MBP, GFAP, and Iba1 analysis, Visiopharm software was used to detect positive staining

based on a threshold of feature band pixel values, creating a project specific configuration. For NeuN quantitation, ROIs were manually drawn in three independent GM regions of the parietal and occipital cortex tissues and subdivided into five tissue layers from each region (Fig. S6b). Feature band Chromaticity Red and FastRed_DAB – Fast Red with a mean 7 × 7 pixel filter was used to detect immunostaining, followed by a polynomial blob filter to identify NeuN-positive cells. For Olig2 quantitation, ROIs were manually drawn around the white matter. Olig2-positive nuclei were detected using feature band HDAB – DAB with a polynomial blob filter and Olig2-negative nuclei were detected using feature band H&E – hematoxylin with a mean 3 ×3 pixel filter and polynomial blob filter. For each analysis, thresholds were adjusted if needed to account for batch-to-batch variability in staining intensity; images were processed in batch mode using the specified configurations to generate the desired outputs.

### Luxol fast blue-PAS-hematoxylin staining

Luxol fast blue (LFB) combined with the periodic acid-Schiff (PAS) procedure was used for histologic examination of white matter from parietal cortex of CTL ($n = 6$) and ZikV ($n = 6$) conditions. On tissue slides, LFB stain highlights the blue myelinated axons of neurons in the white matter tracks and the small dense round nuclei of oligodendrocytes that produce myelin. Demyelination is identified as regions of CNS that lose the blue color that the LFB normally confers to myelin. Fixed tissues 10–15 μm thick were sectioned from paraffin blocks and mounted onto slides. Slides were first deparaffinized and tissue sections hydrated with 95% alcohol. Tissue sections were placed in a tightly capped container with LFB solution at 56 °C overnight. Sections were rinsed in 95% alcohol to remove excess stain followed by rinses in distilled water. Slides were then immersed in lithium carbonate, 0.05% solution for 10–20 sec followed by immersion in 70% alcohol solution until gray and white matter can be distinguished. The sections were then washed in distilled water. The differentiation was finished by rinsing briefly in lithium carbonate solution and then putting through several changes of 70% alcohol solution until white matter sharply contrasted with the gray matter. The sections were thoroughly rinsed in distilled water and placed in 1% periodic acid solution for 5 min followed by rinsing in 2 changes of distilled water. Sections were then placed in Schiff solution for 15 min and washed in tap water for 5 min. Sections were then dehydrated in 95% alcohol and 2 changes of absolute alcohol. The final step was clearing in 3 changes of xylene and mounting with a synthetic resin.

### Magnetic resonance imaging

MRI was performed using a Philips Achieva 3T scanner using acquisition parameters that have been previously described[31]. In brief, a 2D single-shot, half-Fourier turbo spin echo multislice sequence (HASTE) was used to acquire T2-weighted images at various gestational time points (Fig. S1a). Primary analysis of T2 signal abnormality was performed at a single time point representing late gestation for which all animals had imaging (approximately GD120). To quantify the magnitude of the T2 signal abnormality in white matter, a scale from 0–3 was devised, corresponding to normal, mild, moderate, and severe abnormality. A score of 0 reflected the expected signal intensity based on control animals and a score of 3 reflected increased signal intensity matching the signal from surrounding CSF, which was typically the intensity of the primary periventricular lesion in affected animals. Image quality and stability was insufficient to perform analysis of diffusion tensor imaging.

### Electron microscopy analysis

For myelin decompaction analysis, blinded electron microscopy analysis was performed on samples of deep white matter from the mid-parietal cortex (P2 in Fig. S1c). Tissue was fixed in 4% glutaraldehyde in sodium cacodylate buffer at a pH of 7.3, at room temperature, then

stored overnight at 4 °C. The tissue was then washed 5 times in buffer, then post-fixed in 2% buffered osmium tetroxide for 1 h, on ice. The tissue was then washed 5 times in water, dehydrated in a graded series of alcohol, then propylene oxide twice. This was followed by infiltration in 1:1 propylene oxide: epon araldite, 2 changes of epon araldite, and finally polymerization overnight in an oven at 60 °C. Sections of 80 nm thickness were collected on formvar coated slot grids and imaged at 80KV on a JEOL1230 TEM. A formvar grid was used to define consistent regions of analysis in all sections assessed. All axons within five fields of view were analyzed. Ten high-power images (40,000x magnification) within a field of view were generated. ImageJ analysis software was used to measure the axon diameter, fiber diameter (both the axon and myelin sheath) and myelin sheath thickness. The g-ratio was only measured in areas of compact myelin and calculated as the outer diameter across the axon divided by the outer diameter across the axon and myelin sheath measured at the same point.

### Reporting summary

Further information on research design is available in the Nature Portfolio Reporting Summary linked to this article.

### Data availability

The raw sequencing data generated in this study have been deposited in the NCBI Gene Expression Omnibus (GEO) under accession number GSE226401 for the bulk RNA sequencing dataset and GSE227533 for the GeoMx spatial transcriptomic dataset. Source data are provided with this article. Source data are provided with this paper.

### Code availability

The R codes applied to these analyses can be accessed at 10.5281/zenodo.10998903 [https://zenodo.org/records/10998903] and DOI 10.5281/zenodo.10998921 [https://zenodo.org/records/10998921].

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

## Acknowledgements

The authors acknowledge Chris English, Jason Ogle, Audrey Germond, and W. McIntyre Durning at the Washington National Primate Research Center for their contributions to the study execution. We are deeply indebted to the UW Immunology FlaviClub for discussion and suggestions that shaped this study and our interpretations. We thank Jonah Chan (UCSF) for providing input on experiments, Steve Perlmutter (UW) and Philip Horner (Houston Methodist Research Institute) for valuable discussions of the electron microscopy data. We acknowledge Bethany Kondiles (UW) for guidance on electron microscopy analysis and Erica Boldenow (SCRI) for technical support. We appreciate the help of Francisco Perez (UW) in analyzing MRI data. Funding for this study was supported by AI143265 (M.G. and K.A.W.) and AI145296 (M.G.), AI150996 (C.S.), a Core Grant for Vision Research NEI P30EY001730, and by Washington National Primate Research Center grant OD010425. We thank the University of Washington Histology and Imaging Core, Seattle Genomics Core lab, and NanoString Technologies (Seattle, WA) for their services. Funding for Seattle Genomics is supported in part by the National Institutes of Health, Office of the Director P51OD010425 (Seattle MG). Instrumentation used for the GeoMx DSP experiment was supported with Federal funds from the National Institute of Allergy and Infectious Diseases, National Institutes of Health, Department of Health and Human Services, under Contract No. HHSN272201800008C.

## Author contributions

J.T.-G. and C.S. wrote the manuscript, with input from other authors, and coordinated contributions between the collaborating laboratories. M.G. Jr., K.A.W. and L.R. oversaw the NHP model at the Washington National Primate Research Center. RPK performed the fetal brain necropsy and collected samples. K.V. participated in sample collection and processing for RNA analysis. J.S.B. performed the viral qRT-PCR assay. J.M.S. performed the histopathologic assessment of brain tissues. E.P. prepared the specimens for electron microscopy and operated the transmission electron microscope. J.T.-G. collected electron microscopy images and C.S. performed electron microscopy analysis. J.T.-G. and C.S. performed immunohistochemical analyses. C.S. and D.S. performed MRI analysis. J.T.-G. processed samples for transcriptome sequencing. E.S. constructed bulk RNAseq libraries and performed the library sequencing. A.T.G. performed the GeoMx DSP assay on control specimens. L.S.W., D.J.N., and C.J.S. performed the transcriptomic analyses. J.T.-G. and C.S. provided the associated functional interpretation of the transcriptomic datasets. All authors reviewed the final draft of the manuscript.

## Competing interests

The authors declare no competing interests.

## Inclusion & Ethics Statement

All collaborators of this study have fulfilled the criteria for authorship required by Nature Portfolio journals. This study involved collaborations by local researchers throughout the research process, and locally relevant research has been taken into account in citations. Contributor roles and responsibilities were agreed upon ahead of the research that did not result in stigmatization, incrimination, discrimination or personal risk to participants.
