## [Peer Review File · Nature Communications]

REVIEWER COMMENTS

Reviewer #1 (Remarks to the Author):

Reviewer Comments

This manuscript describes disruption in oligodendrocyte maturation and myelin structure in pigtail macaque (*Macaca nemestrina*) fetal brains exposed to zika virus mid to late second trimester of pregnancy. This model has been described previously in two publications by this group using the same set of animals and experimental design (PMID: 27618651 and 29400709). For this study, RNA-seq and Electron microscopy (EM) were added to study the areas shown to have white matter abnormality by MRI from previous studies. The main finding of the study is the downregulation of genes involved in oligodendrocyte maturation and myelin production and myelin decompaction at ultrastructural level in ZIKV exposed fetal brains in otherwise normocephalic fetuses without obvious neuroanatomic defects.

Introduction:

Lines 38-39. The authors cite a limited number of papers for non-human primate models of ZIKV infection but more accurately these are only macaque models of NHP and not representative of NHP models for ZIKV infection and fetal outcomes. Also, they do not note that numerous publications of ZIKV in macaques, even using isolates associated with microcephaly in humans (e.g. French Polynesian, Brazil, Puerto Rican) inoculating in various stages of gestation, were not associated with any fetal or neonatal brain pathology. This impacts the model they describe.

Results:

The quite aggressive (subcutaneous inoculation administered at five different sites on the forearm each with 10^7 plaque forming units (PFU)) and disparate route of inoculation with the two different ZIKV isolates from cell culture (2/6 animals) and mosquito salivary gland extract preparation and a monoclonal dengue virus antibody pre- and post-inoculation (3/6 animals) precludes from making decisive conclusions regarding clinical relevance due to a small sample size, extreme inoculating dose, and the use of one ZIKV isolate (Cambodia) not associated with adverse fetal CNS outcome in humans while the other (Brazil) is associated with CZS and microcephaly. Yet both isolates yielded similar results. The authors need to better explain how a benign ZIKV isolate and contemporary isolate led to similar pathology other than the extreme inoculating dose.

Although the deep white matter area of parietal lobe of ZIKV -exposed fetuses showed downregulation of oligodendrocyte genes responsible for formation and maintenance of myelin sheaths, no difference in staining for Olig2, which stains oligodendrocyte precursors and myelinating oligodendrocytes, was seen between ZIKV and control fetuses. No astrogliosis (GFAP, astrocytes), focal inflammation (Iba-1, microglia) or neuronal loss (NeuN, mature neurons) was observed between the ZIKV and control animals in the parietal and occipital cortex where T2-hyperintense foci

was observed by MRI in the Zikv-exposed animals. The ultrastructural analysis of the axons in the deep white matter area showed normal axonal properties such as no difference in axonal diameter, number of myelin wraps and wrap thickness and no evidence of local inflammation or phagocytosis. In some ZIKV -exposed animals, focal myelin structural disruption described as myelin decompaction was observed. However, no difference in the fraction of the axon diameter composed of myelin (g-ratio) was observed in these animals.

Overall, the ultrastructural data doesn't explain the noticeable T2-weighted MRI signal abnormalities and lesions in the white matter seen in the parietal and occipital regions of the ZIKV -exposed animals. Curiously, although T2-hyperintense foci in the white matter of posterior brain was observed in 4/5 animals using MRI (PMID:29400709), the IHC staining for MBP and the EM data doesn't support any significant damage to the white matter except for in animal Zika 3, which explains the inclusion of data from this animal for Figures 3 and 4. Zika 5 animal despite MRI signal abnormality showed no other anomaly similar to the figures from the control animals. Zika 1 animal featured on their first publication on this model (PMID: 27618651), despite reported significant periventricular lesion on MRI scans, appeared to have no significant white matter injury, tissue and cellular inflammation. EM data from animal is also lacking so it is hard to conclude on abnormalities on axonal myelin structure. It is difficult to assess true damage to the axons based on purely structural disruption without performing functional tests to measure the excitability and conductive capacity of these axons.

In absence of data-supported mechanism to explain the MRI scans, it is difficult to understand the clinical relevance and application of this model.

Discussion:

Line 184: Data is not conclusive to make this statement. Fetuses from two animals inoculated with Cambodian ZIKV isolate were the only ones with detectable ZIKV RNA in the fetal brain. Zika 6 fetal brain also had detectable ZIKV RNA but was excluded from the study due to the use of different set of primers for qPCR detection. It is possible that vertical transmission in Zika 3-5 inoculated with the Brazilian strain at earlier gestation point than the rest of the animals had viral resolution by the end of the study but because of the many different variables introduced to the experimental design of these animals, it is hard to make any conclusion on the fetal outcome of these animals based on the ZIKV strain itself.

Line 187: The disruption of CNS myelin in fetuses is overstated. Zika 3 fetus looks to have the most disruption.

Lines 204-207: This is an unlikely mechanism based on the lack of data showing microglial related inflammation and repair around axons and in the white matter areas.

Lines 213-215: "Hypoxia or infection in utero can cause periventricular leukomalacia (PVL), in which necrotic death of premyelinating oligodendrocytes is accompanied by astrogliosis and microglial activation" In light of this statement, the authors need to discuss a potential role for ZIKV pathology at the level of the placenta. In rhesus macaques, lower dose inoculation using contemporary American isolates of ZIKV noted major placental pathology that could have contributed significantly to findings the authors observed in this study. Further, a placental effect could explain partially how a

benign isolate (Cambodia) can yield significant CNS effects similar to the Brazil isolate- possibly due to the extreme inoculation dose targeting placenta?

Lines 222-229: Data presented doesn't support an inflammatory response to the loss of myelin and myelin decompaction.

Line 236-238: No local neuronal or glial loss was observed to support this statement.

Figure 1 c: The black inset highlighting the alteration in gene expression related to myelination in the deep white matter shows downregulation of these genes in some of the control animals as well. What is the explanation for this?

Figure 2: Information about the number of sections used per animal for each IHC is missing from the Methods and the figure legend. Figure S4 in the figure legend mentions using a single section per animal for quantification. At least 3 sections/animal from different depths of the tissue of interest should be sampled for IHC to make correct quantification.

Figure 3: Where is the data on vacuolization and EM of brain tissue from other ZIKV-exposed animals?

Figure 4: Include images from Zika 5 and 6 showing myelin decompaction. Did Zika 1 and 2 show similar myelin structural disruption. Figure S1 b table shows Zika 6 brain was not used for EM analysis, however, Figure 4 EM data has Zika 6 data. Please correct.

Minor Comment:

Line 169: CSZ should be CZS.

Reviewer #3 (Remarks to the Author):

Thank you for the opportunity to review this very interesting study. It was well written and looked at important aspects of viral injury to the developing brain which may be applicable to other congenital viral infections beyond Zika. I follow a cohort of children with antenatal ZikV infection, so my comments are from the clinical perspective. An animal model study like this allows us to learn so much about the virus and how it may impact the developing brain.

1. Abstract: "Zika virus (ZikV) infection during pregnancy can cause congenital Zika syndrome (CZS) and neurodevelopmental delay in non-microcephalic infants, of which the pathogenesis remains poorly understood." The wording of this sentence is not very clear. Children with and without

microcephaly can have neurodevelopmental delay. Those with microcephaly have what is termed CZS. Are you including non-microcephalic infants without structural brain injury as having CZS?

2. Main lines 23-25 “the pathogenesis of neurodevelopmental delay in CZS displaying normal brain development, termed “normocephalic”, is poorly understood.” CZS is a term used to describe the severe phenotype of congenital Zika infection. I have not seen CZS used to refer to children that are normocephalic and have normal brain development, but developmental delays. Some children with “CZS” may be normocephalic at birth but develop postnatal microcephaly. Their brain imaging is not normal, so for them I feel that using the CZS term is appropriate. As a clinical researcher, the term CZS should be used when referring to children with Zika-associated birth defects or with abnormal structural brain imaging consistent with Zika-infection. Those with just neurodevelopmental delays, should probably not be termed as having “CZS.” See: Characterizing the Pattern of Anomalies in Congenital Zika Syndrome for Pediatric Clinicians - PubMed (nih.gov) Of course, we are continuing to learn about the spectrum of disease, so perhaps the definition of CZS should be expanded. Defining how this term is used in the manuscript is important for the reader to put it in context with human literature.

3. Main line 54- Are the authors proposing an additional feature and definition of CZS?

4. Results Line 70: Did the 6 dams that were viremic, all have fetuses with ZikV RNA present?

5. Results line 82- were transcriptional changes different in the animals that had ZikV RNA at necropsy? It is interesting that only 3 of the fetal brains had ZikV RNA detected.

6. Results line 117- among the 6 ZikV exposed fetuses, was there a difference between the 3 with ZikV RNA in the brain vs. those exposed but without ZikV RNA?

7. In my clinical cohort that I have followed, I have wondered about ZikV infected vs. ZikV exposed. Do the authors consider the fetuses infected or exposed?

8. “Fig. 1. Congenital Zika infection causes downregulation of myelination genes in deep white matter of nonhuman primate”. Should this state Congenital Zika Exposure? Instead of infection? The 3 fetuses of ZIKV infected dams who did not have ZikV RNA detected at necropsy- were they considered ZikV exposed or infected?

9. Fig 2- Is ZIKA 6 represented in panel g and panel i? I do not see that one.

10. Fig 3- Did ZIKA 3 have ZikV RNA present in the brain?

11. Discussion line 180- “neonates” only refers to the first month of age. Since your sentence is regarding motor and cognitive impairment it would be better to use the term “infant” referring to the first year of human life or “child” or “young child”.

12. Discussion line 185- Were all fetal brains normocephalic for gestational age? Based on the Figure showing abnormal T2 hyperintensity, I would anticipate that some of these brains would develop postnatal microcephaly, which has been described in CZS. This may be worth a comment as the trajectory of the brain growth in the study animals postnatally is not known. With the findings, would the authors anticipate the development of microcephaly?

13. Were there subcortical calcifications in any of the brains? Did any of the fetuses have any other features of CZS such as arthrogryposis, eye abnormalities, or were they growth restricted?

14. Were any of the infected dams pregnant with a microcephalic fetus? It seems the study only includes normocephalic fetuses at CS delivery.

15. Can the authors discuss why not all fetuses had ZikV RNA detected in their brain at necropsy. Does the finding of ZikV RNA in the brain make any difference in the impact of the virus on myelin structure and OL maturation?

Reviewer #4 (Remarks to the Author):

In this manuscript, Tisoncik-Go et al. utilized established pigtail macaque fetal Zika virus infection model and uncovered profound disruption of fetal myelin in animals with prenatal ZikV exposure. While the overall research framework is comprehensive, further analysis of specific evidence is needed to enhance its persuasiveness.

Major concerns:

1. The authors claimed that the Zika virus exposed fetuses were non-microcephalic. However, the head circumference data and the diagnostic criteria of microcephalia in pigtail macaque were not mentioned in the text.

2. The inoculation and MRI examination time points illustrated in Fig. S1a varied between maternal animals, as well as the interval between inoculation and cesarean section time, which may bring biases to the downstream analysis.

3. Some data seemed to be contradictory. For example, Fig S1a showed that maternal animals ZIKA 6 and Control 5 were inoculated on gestation day 118 and 134, respectively. However, in Table S1, the inoculation gestational age of ZIKA 6 was day 121 while Control 5 was day 128. The same contradiction could also be seen in Table S2, the age of ZIKA 1, CTL3 and CTL4.

4. Fig S1e showed that, RNA of Zika virus was not found in the brains of fetuses ZIKA 3, 4, 5, and authors did not put forward any other data to prove the fetuses were infected by Zika virus. Whether fetus modeling succeeded remained to be prove.

5. The authors claimed in the abstract that Zika virus exposed animals showed perturbation or remodeling of previously formed myelin. However, the conducted experiments demonstrated a substantial downregulation in gene expression related to crucial components of oligodendrocyte maturation and showcased a disruption in myelination. Notably, there is an absence of evidence

supporting the claim of remodeling of pre-formed myelin. To address this gap, the inclusion of new time points in the experimental design is recommended. This additional temporal dimension will enable a clear differentiation between the remodeling of previously formed myelin and the disruption of myelin formation.

6. Further endeavors in transcriptomic data analysis could be undertaken to elucidate the relationship between neuronal maturation and synaptic formation.

Minor concerns:

1. On the line 70 of the text, authors mentioned that transient viremia was found in 6/7 dams while there were only 6 maternal animals in total and only 5 found virus RNA in plasma. Also, the figure reference should be Fig. S1e rather than Fig. S1d.
2. Fig S1c showed an horizontal brain section, the figure legend annotated it as a coronal plane.
3. On the line 76 of the text, words 'grey matter' should be 'deep grey matter', according to the abbreviation and Fig 1a.

Reviewer #1 (Remarks to the Author):

Reviewer Comments

This manuscript describes disruption in oligodendrocyte maturation and myelin structure in pigtail macaque (*Macaca nemestrina*) fetal brains exposed to zika virus mid to late second trimester of pregnancy. This model has been described previously in two publications by this group using the same set of animals and experimental design (PMID: 27618651 and 29400709). For this study, RNA-seq and Electron microscopy (EM) were added to study the areas shown to have white matter abnormality by MRI from previous studies. The main finding of the study is the downregulation of genes involved in oligodendrocyte maturation and myelin production and myelin decompaction at ultrastructural level in ZIKV exposed fetal brains in otherwise normocephalic fetuses without obvious neuroanatomic defects.

Introduction:

1. **Lines 38-39.** The authors cite a limited number of papers for non-human primate models of ZIKV infection but more accurately these are only macaque models of NHP and not representative of NHP models for ZIKV infection and fetal outcomes.

The reviewer makes an important point that several NHP models of ZIKV infection and fetal outcomes have been reported. In addition to rhesus and pigtail macaque models, we have added citations for marmoset (refs# 27 and 29) and olive baboon (ref# 53) models. These new citations can be found on lines 39 and 212, respectively, of the revised manuscript.

New References added include the following:

27. Seferovic, M. *et al.* Experimental Zika Virus Infection in the Pregnant Common Marmoset Induces Spontaneous Fetal Loss and Neurodevelopmental Abnormalities. *Sci Rep* **8**, 6851, doi:10.1038/s41598-018-25205-1 (2018).
29. Kim, I. J. *et al.* Impact of prior dengue virus infection on Zika virus infection during pregnancy in marmosets. *Sci Transl Med* **15**, eabq6517, doi:10.1126/scitranslmed.abq6517 (2023).
53. Gurung, S. *et al.* Zika virus infection at mid-gestation results in fetal cerebral cortical injury and fetal death in the olive baboon. *PLoS Pathog* **15**, e1007507, doi:10.1371/journal.ppat.1007507 (2019).

2. Also, they do not note that numerous publications of ZIKV in macaques, even using isolates associated with microcephaly in humans (e.g. French Polynesian, Brazil, Puerto Rican) inoculating in various stages of gestation, were not associated with any fetal or neonatal brain pathology. This impacts the model they describe.

We have greatly expanded our discussion of NHP models of CZS to highlight studies that report mild or no histopathologic changes in ZikV-exposed NHP fetuses (refs 24, 76). We also discuss our study limitations, in which we address the applicability of NHP models to human CZS and situate the experimental conditions employed in this study in the broader field of NHP modeling of congenital ZikV infection. This new text can be found on lines 260-284 of the revised manuscript. Addressing both points above, this section of the discussion now reads as follows:

“This study has possible limitations that should be considered in the interpretation of our findings. First, the pathophysiology of congenital ZikV infection in NHP models may not fully replicate that in human CZS. In the current study, ZikV-exposed fetuses did not develop gross microcephaly (>2 s.d. below age-corrected head circumference)³¹, although they did have smaller brain volumes than controls (Table S2), which has been reported in several other NHP models of ZikV infection in pregnancy. Necropsy prior to natural birth in NHP models may obscure the development of secondary microcephaly, whereas

microcephaly in human infants with CZS can progress after birth⁷⁵. While some NHP models of congenital ZikV demonstrate subtle or no neuropathologic changes, most demonstrate gross histopathology that closely mirrors human CZS⁷⁷. Second, we were unable to verify fetal infection by PCR in 3/6 ZikV-exposed fetuses, although we confirmed maternal infection in all cases. These observations mirror human data in which ZikV RNA is detectable in less than half of CZS cases at birth. We have proposed a brain-intrinsic mechanism for myelin perturbation, but these experiments do not definitively rule out the possibility of extra-fetal mechanisms including maternal inflammatory cytokines or placental disruption. Although chronic placental inflammation has been associated with white matter injury in premature infants, the only placental pathology observed in ZikV-exposed animals was mild deciduitis, which was also present in some control animals. Third, we present a cohort of 6 ZikV-exposed and 6 control fetuses in which only one animal from each experimental condition was male. This may have precluded our ability to detect subtle neuropathology and/or sex-related changes. Finally, we combined analysis of animals infected with two closely-related isolates of ZikV and across a range of gestational ages from first and second trimester. We did not detect strain-related differences on sub-group analysis. Although the study may not have been sufficiently powered to detect small differences, the recapitulation of myelin decompaction in both strains and across time points argues for a conserved pathophysiologic mechanism and suggests that white matter injury may be common in human CZS.”

Results:

3. The quite aggressive (subcutaneous inoculation administered at five different sites on the forearm each with 10^7 plaque forming units (PFU)) and disparate route of inoculation with the two different ZIKV isolates from cell culture (2/6 animals) and mosquito salivary gland extract preparation and a monoclonal dengue virus antibody pre- and post-inoculation (3/6 animals) precludes from making decisive conclusions regarding clinical relevance due to a small sample size, extreme inoculating dose, and the use of one ZIKV isolate (Cambodia) not associated with adverse fetal CNS outcome in humans while the other (Brazil) is associated with CZS and microcephaly. Yet both isolates yielded similar results.

The inoculation dose used in his study was based on published estimates of the infectious dose transmitted by an infected mosquito (Styer et al., PLoS Pathogens 2007, PMID: 17941708; Li et al, PLoS Neglected Tropical Diseases, PMID: 22953014), which range from 10^2 to 10^8 infectious particles per bite, and are repeated as many as 10 times with serial biting. We acknowledge that the inoculation dose used in this study is at the higher end of that range, and higher than those used in many NHP studies. However, the resulting peak maternal viremia we observed (ranging from 10^2 to 10^6 copies/mL) lies entirely within the range described in other published studies of pregnant macaque models of ZikV infection, and our observation of clinical symptoms and distribution of viral RNA also matched published results (see Adams Waldorf et al., Nat. Med. 2018 as compared to, e.g., Hirsch et al., PLoS Pathogens 2017, PMID: 28278237 or Martinot et al., Cell 2018).

We appreciate the Reviewer’s comment that strain-specific differences and variability in the inoculum formulation are not well addressed in our study due to the small sample size; we have performed additional analysis and introduced text in the discussion section to address the limitations. We performed post-hoc analysis of our results for spatial transcriptomics and immunohistochemistry considering viral strain as an independent variable and did not find significant differences between the two isolates. We also note that although the 2010 Cambodian isolate we used here has not been directly implicated in CZS, it is closely related to Asian-lineage strains with presumed endemic circulation in Southeast Asia that have been associated with human microcephaly (see Wongsurawat et al., Emerg. Infect Dis. 2018, PMID: 29985788). Because we observed myelin perturbation with both viral isolates as the reviewer points out, we feel this in fact represents a strength of our study, as it demonstrates a

mechanism that is likely conserved across both strains of virus and may therefore affect children in geographically diverse areas who were exposed to ZikV *in utero*.

We do not directly assess difference in outcome related to the presence or absence of Dengue virus antibody treatment, as (1) this was assessed in a previous publication (Adams Waldorf et al., Nat Med, 2018) and (2) we did not observe any differences in myelination phenotype according to Dengue antibody exposure.

4. The authors need to better explain how a benign ZIKV isolate and contemporary isolate led to similar pathology other than the extreme inoculating dose.

We hypothesize that ZikV strains from Asian and American lineages are sufficiently similar to have largely overlapping pathophysiology in congenital infection. In contrast, African-lineage ZikV strains typically cause more severe pathogenesis, trigger a stronger immune response, and are thought to result in outright fetal demise as opposed to congenital anomalies (Rosinski et al., PLoS Pathogens 2023, PMID: 36976812 and Aubry et al., Nat. Comm. 2021, PMID: 33568638). The two strains utilized in our study share 99.4% sequence identity at the nucleic acid level, but only share 88.7% sequence identity with the prototypical African-lineage strain from Uganda (Esser-Nobis et al., J. Virol. 2019 PMID: 31019057). We acknowledge that two notable studies in mice have identified differences in neuropathology of Asian- and American-lineage ZikV (Yuan et al., Science 2017, PMID: 28971967 and Zhang et al., EBioMed. 2017, PMID: 29107512); however, subsequent data from Thailand have shown that Asian-lineage viruses can cause microcephaly in humans (Wongsurawat et al., Emerg. Infect Dis. 2018, PMID: 29985788. Our interpretation of these studies is that the pathophysiology of CZS is multifactorial and did not arise as a consequence of a single, or even several, mutations in the ZikV genome. We hypothesize that, while Asian-lineage ZikV may cause a milder CZS phenotype than American-lineage ZikV and may be less likely to cause overt microcephaly, the two lineages share sufficient sequence identity to cause similar neuropathology in fetal macaques. Thus, fetal infection by either lineage warrants close monitoring and evaluation. Indeed, this hypothesis is supported by the paper from Zhang et al., EBioMed. 2017 (PMID: 29107512), in which both the Cambodian strain we used and a Venezuelan isolate from 2016 cause microcephaly in mice, including a decrease in cortical MBP expression, although the Venezuelan strain produces a more severe phenotype.

5. Although the deep white matter area of parietal lobe of ZIKV -exposed fetuses showed downregulation of oligodendrocyte genes responsible for formation and maintenance of myelin sheaths, no difference in staining for Olig2, which stains oligodendrocyte precursors and myelinating oligodendrocytes, was seen between ZIKV and control fetuses. No astrogliosis (GFAP, astrocytes), focal inflammation (Iba-1, microglia) or neuronal loss (NeuN, mature neurons) was observed between the ZIKV and control animals in the parietal and occipital cortex where T2-hyperintense foci was observed by MRI in the Zikv-exposed animals. The ultrastructural analysis of the axons in the deep white matter area showed normal axonal properties such as no difference in axonal diameter, number of myelin wraps and wrap thickness and no evidence of local inflammation or phagocytosis. In some ZIKV -exposed animals, focal myelin structural disruption described as myelin decompaction was observed. However, no difference in the fraction of the axon diameter composed of myelin (g-ratio) was observed in these animals.

The small number of animals and the interval between maternal inoculation and fetal necropsy (ranged between 23 and 97 days post-inoculation) is a potentially important variable that may explain the lack of differences the Reviewer notes, and we now acknowledge this limitation in a new discussion paragraph, as noted above. While we do not see a significant difference between control and ZikV-exposed fetuses for Iba1 signal in total brain area quantified from parietal cortex, there are focal increases in Iba1 signal in white matter and adjacent to the ventricle for some ZikV animals. We have included new

micrographs from ZIKA4 to provide additional evidence of these changes (Fig. S5f). It is likely that any microglial response would be transient and peak closer to the time of peak viral replication in the brain, which may be many weeks prior to necropsy that was conducted close to term gestation (Ave GD151). We also acknowledge the electron microscopy data involved a small sample number due to limited tissue availability, and this may have precluded our ability to detect subtle changes in axon or myelin ultrastructure. The g-ratio estimates and axon diameter shown in Fig S7d-e could reflect experimental variability (e.g. due to sampling location or gestational age), or a fundamental feature of pathophysiology.

6. Overall, the ultrastructural data doesn't explain the noticeable T2-weighted MRI signal abnormalities and lesions in the white matter seen in the parietal and occipital regions of the ZIKV -exposed animals. Curiously, although T2-hyperintense foci in the white matter of posterior brain was observed in 4/5 animals using MRI (PMID:29400709), the IHC staining for MBP and the EM data doesn't support any significant damage to the white matter except for in animal Zika 3, which explains the inclusion of data from this animal for Figures 3 and 4. Zika 5 animal despite MRI signal abnormality showed no other anomaly similar to the figures from the control animals.

The Reviewer astutely observes that the T2-weighted signal abnormality in the deep white matter is out of proportion to the degree of direct cellular damage or inflammation we measured in ZIKV-exposed fetal brains. Several features of the MRI findings should be considered in interpreting our findings. First, it should be noted that MR imaging is relatively nonspecific and limited data exist directly correlating MRI findings with neuropathology in fetal white matter. Second, in order to age-match ZikV and control animals, we compared imaging at around 120 days of gestation, and the underlying process reflected in this imaging may have evolved significantly during the ~30 days between imaging and necropsy. Third, the MRI findings demonstrate some regional heterogeneity and this may have led to our detection of severe phenotypes in only some animals at some locations. Despite these caveats, the ultrastructural phenotype of myelin decompaction was shared across all animals evaluated. We have now included new EM micrographs from ZIKA5 and ZIKA6 to provide additional evidence of this phenotype across animals (Fig. S7a). Furthermore, because myelin decompaction can be explained as a consequence of the gene expression changes and MBP protein loss identified in transcriptomic and immunohistochemical analyses, we hypothesize that decompaction is the common process reflected in the MRI data.

7. Zika 1 animal featured on their first publication on this model (PMID: 27618651), despite reported significant periventricular lesion on MRI scans, appeared to have no significant white matter injury, tissue and cellular inflammation. EM data from animal is also lacking so it is hard to conclude on abnormalities on axonal myelin structure. It is difficult to assess true damage to the axons based on purely structural disruption without performing functional tests to measure the excitability and conductive capacity of these axons.

We previously reported on the white matter injury of ZIKA1 that included the presence of gliosis with apoptotic features, increased enrichment of GFAP-positive reactive astrocytes, and importantly, structures that are consistent with axonal spheroids and a pathologic sign of axon injury (Adams Waldorf et al., Nat Med 2016). EM samples were not collected for ZIKA1; therefore, as the Reviewer notes, we are unable to conclude abnormalities on axonal myelin structure for this animal. We agree it would be informative to do functional tests; however, this is not a possibility for ZIKA1 or any of the other animals reported here, as they have been processed through necropsy including fixation in formalin.

8. In absence of data-supported mechanism to explain the MRI scans, it is difficult to understand the clinical relevance and application of this model.

Our study adds to the collective findings that define fetal neuropathological profiles of brain injury underlying congenital Zika syndrome resulting from Zika virus infection during pregnancy for direct translation to human disease. The strength of our study lies in the observation of a myelin decompaction phenotype in all of the ZikV-exposed fetuses we examined, and identification of ZikV-related perturbations in gene expression for oligodendrocyte development and myelin production as well as neuronal development and synaptogenesis. We acknowledge that our findings do not identify a unifying mechanism, and this is an important area for future work to prevent or reverse white matter injury in CZS. An major clinical implication of these findings is the importance of evaluating ZikV-exposed infants for white matter injury. Even in the absence of targeted pharmacotherapy to reverse white matter injury, our findings suggest that rehabilitation and physical therapy focusing on recovering from myelin damage may play an important role in CZS.

Discussion:

9. **Line 184:** Data is not conclusive to make this statement. Fetuses from two animals inoculated with Cambodian ZIKV isolate were the only ones with detectable ZIKV RNA in the fetal brain. Zika 6 fetal brain also had detectable ZIKV RNA but was excluded from the study due to the use of different set of primers for qPCR detection. It is possible that vertical transmission in Zika 3-5 inoculated with the Brazilian strain at earlier gestation point than the rest of the animals had viral resolution by the end of the study but because of the many different variables introduced to the experimental design of these animals, it is hard to make any conclusion on the fetal outcome of these animals based on the ZIKV strain itself.

The Reviewer's concern relates to a lack of convincing evidence that fetal infection with ZikV underlies the white matter injury we observed, which is a valid concern considering the different variables of the experimental design. It is possible that the different gestational ages at ZikV inoculation may explain why virus is not detected in the fetal brain of all animals. We agree that ZikV likely cleared the fetal brain in ZIKA3-5 due to an earlier inoculation at GD60-63, as compared to ZIKA1-2 that were inoculated later in gestation (GD119 and GD82, respectively). To avoid overstating our findings, we have elected to refer to all fetuses as "ZikV-exposed", as we confirmed maternal infection for all 6 dams. We hypothesize that fetal infection occurred in all cases and favor a mechanism involving primary infection of the fetal brain, but we acknowledge that other potential mechanisms, including disrupted placental function, are possible explanations. We acknowledge that, with only 3/6 ZikV-exposed fetuses demonstrating viral RNA, we cannot definitively conclude that the myelin phenotype we observe is due to viral infection in the fetal brain. We have adjusted the text of the paper to reflect this uncertainty, and to point out several lines of evidence supporting fetal viral infection, as outlined below.

- First, we have added text in the discussion (lines 205-209) clarifying that 2/3 fetuses in which we did not detect viral RNA had MRI findings of a primary lesion in the neural progenitor niche, which we hypothesize reflects active infection in that area.

"We were able to directly confirm ZikV infection in 3/6 fetuses, while two of three fetuses in which we did not detect ZikV RNA had MRI findings demonstrating a "primary" lesion in the posterior periventricular region, the niche of neural progenitor cells (NPCs), arguing that they were infected with ZikV but fetal brain infection was cleared at the time of our analysis."

- Second, we have added text in the supplemental figure legend of Fig. S1 (lines 45-46), indicating that the three animals without detectable ZikV RNA also had the longest interval after inoculation prior to necropsy. As the reviewer points out, this may have allowed the animals to clear the virus, which appears to be the most common outcome of human fetal infection, both in fetuses that develop microcephaly and those that do not (see Oliveira et al., Int J. Gyn. Ob 2020, PMID: 31975394).

- Third, we have clarified that an important alternate explanation that we cannot rule out is the possibility of disrupted placental physiology, which might be expected to result in fetal hypoxia or other features of placental insufficiency. We have added the following sentences in the discussion (lines 271-276) addressing this possibility:

“We have proposed a brain-intrinsic mechanism for myelin perturbation, but these experiments do not definitively rule out the possibility of extra-fetal mechanisms including maternal inflammatory cytokines or placental disruption. Although chronic placental inflammation has been associated with white matter injury in premature infants, the only placental pathology observed in ZikV-exposed animals was mild deciduitis, which was also present in some control animals.” (Adams Waldorf et al., Nat Med, 2018; Fig. S15).

- Fourth, we have performed additional sub-group analyses of the transcriptomic and immunohistochemical datasets to assess whether any differences could be detected between animals with and without detectable ZikV RNA. The following lines have been added in the text, and corresponding figures have been updated:

Lines 89-91: “Principal component analysis did not reveal differences between animals according to detection of ZikV RNA in fetal tissue or ZikV strain inoculated, but rather tissue region was the greatest source of variation (Fig. S2d).”

Lines 136-138: “There were no significant differences in immunohistochemical quantification of MBP, GFAP, or Iba1 when comparing between ZikV-exposed animals based on detection of ZikV RNA in fetal tissue or ZikV strain inoculated (Fig. S1e).”

10. **Line 187:** The disruption of CNS myelin in fetuses is overstated. Zika 3 fetus looks to have the most disruption.

All ZikV-exposed fetuses with tissue examined using electron microscopy exhibited myelin decompaction (Fig. 4e). To address this within the body of the paper, we have added more EM images from additional animals in Figure S7a demonstrating the decompaction phenotype. Only 3/6 ZikV-exposed fetuses had tissue collected for electron microscopy and we have included EM images from all animals with tissue available.

11. **Lines 204-207:** This is an unlikely mechanism based on the lack of data showing microglial related inflammation and repair around axons and in the white matter areas.

We agree with this comment by the Reviewer and have modified text to clarify that we did not see evidence for OPC expansion or microglial-driven repair mechanisms. The revised text (lines 217-221) reads as follows:

“Here, fetal exposure to ZikV did not result in changes in the density of Olig2+ cells in white matter. In some conditions of neurologic injury, Olig2+ OPC populations expand and differentiate to oligodendrocytes, and facilitating repair and re-myelination of injured axons. However, expansion of OPCs in these instances leads to upregulation of Olig2 and other early markers of the oligodendrocyte lineage.”

12. **Lines 213-215:** “Hypoxia or infection in utero can cause periventricular leukomalacia (PVL), in which necrotic death of premyelinating oligodendrocytes is accompanied by astrogliosis and microglial activation” In light of this statement, the authors need to discuss a potential role for ZIKV pathology at the level of the placenta. In rhesus macaques, lower dose inoculation using contemporary American isolates of ZIKV noted major placental pathology that could have contributed significantly to findings the

authors observed in this study. Further, a placental effect could explain partially how a benign isolate (Cambodia) can yield significant CNS effects similar to the Brazil isolate- possibly due to the extreme inoculation dose targeting placenta?

The Reviewer makes an important point that placental pathology could have contributed to the white matter injury we describe. As placental driven changes could be a mechanism and we cannot exclude a primary placental abnormality with secondary (e.g. hypoxic) injury, we have added the following sentences to the discussion section to elaborate on this point (lines 271-283):

“We have proposed a brain-intrinsic mechanism for myelin perturbation, but these experiments do not definitively rule out the possibility of extra-fetal mechanisms including maternal inflammatory cytokines or placental disruption. Although chronic placental inflammation has been associated with white matter injury in premature infants, the only placental pathology observed in ZikV-exposed animals was mild deciduitis, which was also present in some control animals.”

13. **Lines 222-229:** Data presented doesn't support an inflammatory response to the loss of myelin and myelin decompaction.

We agree with the Reviewer's comment and we do not see evidence of inflammation or immune infiltrate to explain the loss of myelin. We have rephrased the sentence (lines 235-239) to clarify this point as follows:

“However, in our study there was no histopathologic evidence of inflammatory infiltrate and minimal induction of proinflammatory pathways from the transcriptomic data in the fetal brain of the ZikV cohort animals, indicating the demyelinating phenotype is not the result of a chronic T cell-mediated autoimmune inflammatory response against myelin proteins or phagocytic attack of oligodendrocytes.”

14. **Line 236-238:** No local neuronal or glial loss was observed to support this statement.

The Reviewer is correct in that we do not see any evidence of either necrosis or increased apoptosis in H&E-stained sections. We agree that this is a speculative statement and we have rephrased this sentence accordingly (lines 249-251) to read as follows:

“Therefore, we propose that the disruption of myelin may be related to a loss of trophic or maturation signals derived from local neurons or even astrocytes.”

15. **Figure 1 c:** The black inset highlighting the alteration in gene expression related to myelination in the deep white matter shows downregulation of these genes in some of the control animals as well. What is the explanation for this?

The pattern within the inset of Fig. 1c noted by the Reviewer is likely the result of subtle differences in the tissue processing. These differences along with variation that is inherent to the necropsy process likely explain the pattern seen here. Indeed, a close inspection of the heatmap demonstrates that three of the four control samples with lower expression of myelin genes represent a single animal (CTL 1), and these samples also have lower expression of astrocyte and neuron genes in the rows above, suggesting that processing of this animal's tissue may have led to some systematic difference in gene counts. The analysis used for differential gene expression in the GeoMx data uses a linear mixed effects model, which is specifically designed to account for this type of inter-animal variation. Notably, we analyzed the data using both conventional differential gene expression with the linear mixed model, and the reduction in myelin genes was highly significant.

16. **Figure 2:** Information about the number of sections used per animal for each IHC is missing from the Methods and the figure legend. Figure S4 in the figure legend mentions using a single section per animal for quantification. At least 3 sections/animal from different depths of the tissue of interest should be sampled for IHC to make correct quantification.

We appreciate the Reviewer's concern regarding variability in IHC data and the need for sufficient replicates to confirm findings using this technique. To address this, we have performed a new immunohistochemical analysis of MBP expression in white matter, adding new staining of sections for animals where tissue was available. These new analyses provide additional evidence of myelin perturbation throughout the cortex.

We have added additional paragraphs in the Methods section describing the fetal brain sampling scheme (lines 394-403) and expanded the description of the IHC quantification (lines 549-571).

It is important to acknowledge that given the large number of parallel assays performed on the same tissue, we were unable to perform immunohistochemistry on slices from precisely the same brain area for all animals and all stains. To compensate for this, we refined our analysis to identify three subregions of white matter for analysis (Fig. S4a). Next, we selected two regions that had location-matched sections across the most ZikV and CTL animals and included these sites in the primary statistical analysis (Fig. 2g-h; superior gyri from parietal cortex and inferior gyri of occipital cortex). For transparency, we added additional figure panels (Fig. S4b-d) showing the MBP quantification within each of these three regions across all available sections, which clearly demonstrate the reproducibility of reduced staining for MBP across the ZikV-exposed fetal brains. With these additional data, our IHC analysis now includes a total of 18 images representing 17 locations from CTL animals and 20 images representing 19 locations from ZikV-exposed animals.

We would also like to reiterate that we chose to perform luxol fast blue staining to provide corroboration of our IHC findings in a related assay.

In addition to the changes above to figures and analysis, we have included the following additions to the methods section to describe this technique (lines 551-556):

“For MBP quantitation, ROIs were manually drawn around subcortical deep white matter contained within superior gyri, inferior gyri, or deep projection tracts (Fig. S4a). Slice location along the rostro-caudal axis was identified from reference images of *Macaca mulatta* available in the Scalable Brain Atlas (<https://scalablebrainatlas.incf.org/macaque/DB09>). For primary statistical analysis (Fig. 2g-h) sections were only included if they matched rostro-caudal and ROI location across animals, and had been stained in parallel.”

Additional information is provided in the legend for Fig. S4 (lines 103-112):

“**a**) section of occipital cortex (from CTL 3) stained for MBP highlighting the regions of interest (ROIs) in white matter for which MBP staining was quantified, representing subcortical WM in superior gyri (green), inferior gyri (blue) and central deep WM tracts (red). **b-d**) Quantification of MBP staining area in the WM from **b**) superior gyri, **c**) deep tract, and **d**) inferior gyri, measured as the ratio of area occupied by chromogen divided by the total area of the ROI. AP Location refers to distance from the anterior commissure along the rostro-caudal axis (negative values are more caudal) referenced to the adult *Macaque Scalable Brain Atlas* (see Methods). Each point represents an individual section; for control, n=6 animals, 17 tissue blocks; for ZikV, n=6 animals, 19 tissue blocks. Black line, linear regression of CTL points; grey line, linear regression of ZikV points; shaded areas, 95% confidence interval.”

17. **Figure 3:** Where is the data on vacuolization and EM of brain tissue from other ZIKV-exposed animals?

We thank the Reviewer for their comment and agree that it's important to report the data on vacuolization and EM of brain tissue for all ZikV animals where available. To address this comment, we have performed a new semi-quantitative analysis of vacuolar changes in four different regions of white matter across all controls and ZikV animals where H&E-stained slides were available. New text has been added to the manuscript (including a description in the Methods, lines 513-521) along with a new panel e in Fig. 3 reporting the vacuolation score in frontal, parietal, corpus callosum, and occipital regions of cortex. We have moved the original pane e from Fig. 3 to Suppl Fig 5g and have included additional EM images of ZIKA5 and ZIKA6 parietal and occipital tissues where available. The revised text (lines 149-155) reads as follows:

“On hematoxylin and eosin (H&E) staining of white matter, we did not observe any evidence of inflammatory infiltrate. While we noted vacuolar changes in the deep white matter of both groups, there was a trend toward more severe vacuolation in the white matter of ZikV-exposed animals than controls (**Fig. 3d-e**). In the DGM overlying the site of the primary periventricular lesion, EM revealed variable disruption of the brain parenchyma that was not observed in control animals, while in parietal grey matter there were less severe changes to ultrastructural architecture (**Fig. S5g**).”

18. **Figure 4:** Include images from Zika 5 and 6 showing myelin decompaction.

We have added EM images of ZIKA 5 and ZIKA6 white matter to Fig. S7a demonstrating the decompaction phenotype in these animals. EM images from an additional control animal (CTL 5) is included in Fig. 7a for comparison. We have added new text to the figure legend of Fig. S7 on the new EM images in panel a. We have also added additional text to the figure legend of Fig. S7 on panels b and c to provide more detail of the analysis shown in each panel. The new text (lines 152-158) is as follows:

“**a)** top row, representative EM images of deep white matter from CTL 5, ZIKA 5 and ZIKA 6 animals. Inset in the bottom row shows high-magnification (40,000x) images of the area marked by the yellow rectangle, which are representative of the images used for quantification. **b)** quantification of the number (top) or fraction (bottom) of large-diameter (>250 nm) axons with mature myelin sheaths. **c)** analysis of the fraction of axons myelinated according to gestational age (top row) or days post-ZikV inoculation (bottom row).”

19. Did Zika 1 and 2 show similar myelin structural disruption. Figure S1 b table shows Zika 6 brain was not used for EM analysis, however, Figure 4 EM data has Zika 6 data. Please correct.

We apologize for the mistake—in the creation of the table, we mistakenly reversed the EM data sample assignments for ZikV-exposed and control animals. We thank the Reviewer for their careful reading and catching this error. The table in Figure S1 has now been corrected.

Minor Comment:

20. **Line 169:** CSZ should be CZS.

We have changed this sentence in the updated manuscript to the following:

“Overall, there were no differences in fetal disease phenotype across ZikV animals following exposure to either ZikV strain used in our studies, showing that myelin perturbation is not strain specific for Asian lineage ZikV.”

Reviewer #3 (Remarks to the Author):

Thank you for the opportunity to review this very interesting study. It was well written and looked at important aspects of viral injury to the developing brain which may be applicable to other congenital viral infections beyond Zika. I follow a cohort of children with antenatal ZikV infection, so my comments are from the clinical perspective. An animal model study like this allows us to learn so much about the virus and how it may impact the developing brain.

1. **Abstract:** “Zika virus (ZikV) infection during pregnancy can cause congenital Zika syndrome (CZS) and neurodevelopmental delay in non-microcephalic infants, of which the pathogenesis remains poorly understood.” The wording of this sentence is not very clear. Children with and without microcephaly can have neurodevelopmental delay. Those with microcephaly have what is termed CZS. Are you included non-microcephalic infants without structural brain injury as having CZS?

We thank the Reviewer for their comment and have revised the sentence in the abstract to clarify that CZS will be used to describe only infants with abnormal physical exam findings. The revised sentence (lines 1-2) reads as follows:

“Zika virus (ZikV) infection during pregnancy can cause congenital Zika syndrome (CZS) and neurodevelopmental delay in infants, of which the pathogenesis remains poorly understood.”

We have also updated our discussion to emphasize that our results argue that ZIKV-exposed infants without overt CZS may need careful evaluation for developmental delay or other consequences of myelin disruption.

2. Main lines 23-25 “the pathogenesis of neurodevelopmental delay in CZS displaying normal brain development, termed “normocephalic”, is poorly understood.” CZS is a term used to describe the severe phenotype of congenital Zika infection. I have not seen CZS used to refer to children that are normocephalic and have normal brain development, but developmental delays. Some children with “CZS” may be normocephalic at birth but develop postnatal microcephaly. Their brain imaging is not normal, so for them I feel that using the CZS term is appropriate. As a clinical researcher, the term CZS should be used when referring to children with Zika-associated birth defects or with abnormal structural brain imaging consistent with Zika-infection. Those with just neurodevelopmental delays, should probably not be termed as having “CZS.” See: Characterizing the Pattern of Anomalies in Congenital Zika Syndrome for Pediatric Clinicians - PubMed (nih.gov) Of course, we are continuing to learn about the spectrum of disease, so perhaps the definition of CZS should be expanded. Defining how this term is used in the manuscript is important for the reader to put it in context with human literature.

We agree with the Reviewer that the definition of CZS continues to evolve as additional insight is gained from cohort studies of children exposed to ZikV *in utero*. We do not propose a fundamental change to the definition of congenital Zika syndrome, as outlined in the paper referenced by the reviewer (Moore et al., PMID). As that paper states, “although the numbers are small, recent reports provide evidence that the distinctive brain and eye anomalies of congenital ZikV infection can occur without microcephaly”. In our study, the majority of ZikV-exposed NHP fetuses had abnormal MRI signal, which we believe would meet criteria for what the reviewer calls “structural brain injury” if found in ZikV-exposed infants. Several studies following cohorts of non-microcephalic children with *in utero* ZikV exposure have documented neurodevelopmental delay, but many of these children have not had MRI-based neuroimaging. In studies where neuroimaging has been performed, non-microcephalic infants CZS can have more subtle structural abnormalities, which include delayed myelination (e.g., Arago et al., AJNR 2017, PMID: 28522665). We feel that our results can be situated within this literature as providing additional evidence that white

matter injury is part of CZS and yielding insight into possible mechanisms of myelin perturbation seen in infants with CZS.

We have made several changes to the text to clarify this position, as follows:

- In the introduction, we have adjusted the phrasing to reflect the distinctions described above. The new sentence (lines 23-25) reads as:
“The mechanism of microcephaly in CZS is thought to be related to ZikV infection and death of neural progenitor cells leading to decreased neurogenesis. However, the pathogenesis of neurodevelopmental delay in CZS, particularly in those displaying normal head circumference, termed “normocephalic”, is poorly understood.”
- In the discussion, we have updated the phrasing to contextualize our findings within the broader literature of CZS. The new sentence (lines 187-190) reads as:
“While microcephaly is a hallmark of severe CZS, additional neuroanatomic abnormalities have been defined using neuroimaging, including reports of delayed myelination. Moreover, normocephalic infants with *in utero* ZikV exposure may have higher rates of neurodevelopmental delay.”

3. **Main line 54-** Are the authors proposing an additional feature and definition of CZS?

We do not propose a fundamental change to the definition of congenital Zika syndrome, but we would expect that infants with similar findings to those we describe would be characterized as having CZS. We acknowledge that our findings are characterized in a macaque model and may not entirely recapitulate CZS in humans. However, our work provides novel insight into the mechanisms of CZS beyond cell death in neural progenitors and cortical volume loss. In addition, our findings suggest that infants without microcephaly may have abnormalities in white matter development or function, which may offer clinicians a mechanism for diagnosing CZS and/or therapeutic targets for improving outcomes in CZS. We elected to take a conservative approach and did not call for a revision of CZS diagnostic criteria to provide more focus to white matter changes, but we hope that our findings guide future studies, which may evaluate whether this should be the case.

We have altered this line in the manuscript to reflect the evolving definition of congenital Zika syndrome (lines 52-54); it now reads as follows: “These findings argue that oligodendrocyte alteration leading to dysregulation of myelination and myelin wrap maintenance are features of CZS.”

4. **Results Line 70:** Did the 6 dams that were viremic, all have fetuses with ZikV RNA present?

Not all the fetuses had detectable RNA. Dams (ZIKA 3-5) were viremic at 2 days post-inoculation, but the fetuses had undetectable viral RNA in tissue at the time of necropsy. One dam (ZIKA6) had undetectable viremia, and ZikV RNA was detected in fetal tissue at the time of necropsy. As necropsy was performed for ZIKA3-5 at least 60 days after maternal ZikV inoculation, we hypothesize that they may have cleared the virus by the time of necropsy. ZIKA 1, 2 and 6 were inoculated at later gestational ages and necropsy was performed after a shorter time interval, which is likely why ZikV RNA was detected in fetal tissue for these animals.

5. **Results line 82-** were transcriptional changes different in the animals that had ZikV RNA at necropsy?

It is interesting that only 3 of the fetal brains had ZikV RNA detected.

The Reviewer raises an important question and we have performed additional sub-group analysis of the transcriptomic data to assess whether animals with and without detectable ZikV RNA were

significantly different. The Principal Component Analysis (PCA) of the whole transcriptome for the spatial dataset shows sample clustering by tissue region and not by ZikV RNA detection along PC1. This demonstrates tissue region as the main driver of transcriptomic variation in the fetal brain, with ZIKA 1 and ZIKA 6 showing the greatest separation along PC2. This new analysis shows that there are no significant differences between animals based on the presence of ZikV RNA at necropsy. These results are included in a new panel d of Figure S2. The new text (lines 89-91) is as follows:

“Principal component analysis did not reveal differences between animals according to detection of ZikV RNA in fetal tissue or ZikV strain inoculated (Fig. S2d).”

6. **Results line 117-** among the 6 ZikV exposed fetuses, was there a difference between the 3 with ZikV RNA in the brain vs. those exposed but without ZikV RNA?

Please see our response to the question above on the new sub-group analysis we performed on the transcriptomic dataset. To fully address this question, we have also performed a new sub-group analysis of the immunohistochemical data to determine whether any differences could be noted between fetuses with and without detectable ZikV RNA at necropsy. There were no significant differences between animals based on ZikV RNA presence at necropsy ($p=0.72$ for parietal sections and $p=0.67$ for occipital Welch's t-test).

7. In my clinical cohort that I have followed, I have wondered about ZikV infected vs. ZikV exposed. Do the authors consider the fetuses infected or exposed?

We can confirm that ZIKA1-2 and 6 fetuses were infected, but we only have circumstantial evidence supporting fetal infection in ZIKA3-5. ZikV infection is based on detection of viral RNA in fetal tissues at necropsy by qRT-PCR assay. To avoid overstating our results we have elected to refer to all fetuses as “ZikV-exposed,” grouping them together as the myelin perturbation phenotype appears conserved across all animals in our study. We have clarified this distinction throughout the manuscript, including the following sentences (lines 205-209) in the discussion:

“We were able to directly confirm ZikV infection in 3/6 fetuses, while two of three fetuses in which we did not detect ZikV RNA had MRI findings demonstrating a “primary” lesion in the posterior periventricular region, the niche of neural progenitor cells (NPCs), arguing that they were infected with ZikV but fetal brain infection was cleared at the time of our analysis.”

8. **“Fig. 1. Congenital Zika infection causes downregulation of myelination genes in deep white matter of nonhuman primate”.** Should this state Congenital Zika Exposure? Instead of infection? The 3 fetuses of ZIKV infected dams who did not have ZikV RNA detected at necropsy- were they considered ZikV exposed or infected?

Please see our response to Q7 above. Based on our reasoning above, we have corrected Fig. 1 legend title to read **“Fetal ZikV exposure causes downregulation of myelination genes in deep white matter of nonhuman primate”**. This revision can be found on line 298 of the revised text.

9. **Fig 2-** Is ZIKA 6 represented in panel g and panel i? I do not see that one.

We appreciate the Reviewer's keen attention to this detail. Unfortunately, we did not have location-matched tissue available to perform immunohistochemical analysis on parietal cortex for ZIKA6. Due to the large number of different assays performed on the same fetal brain samples, in some instances we did not have sufficient tissue remaining to perform location-matched replicates of certain assays. In order

to provide transparency regarding this limitation we made the following changes to the revised manuscript

- We have included a table in Fig. S1 outlining the tissue analyzed for each animal.
- We have updated individual figure legends and the results section for each assay to indicate the number of animals included for analysis.

10. **Fig 3-** Did ZIKA 3 have ZikV RNA present in the brain?

ZIKA3 did not have detectable ZIKV RNA in the fetal brain based on a ZIKV-specific qRT-PCR assay.

11. **Discussion line 180-** “neonates” only refers to the first month of age. Since your sentence is regarding motor and cognitive impairment it would be better to use the term “infant” referring to the first year of human life or “child” or “young child”.

We appreciate the distinction made by the Reviewer and have changed this phrasing to “developing children” on line 187 of the revised manuscript.

12. **Discussion line 185-** Were all fetal brains normocephalic for gestational age? Based on the Figure showing abnormal T2 hyperintensity, I would anticipate that some of these brains would develop postnatal microcephaly, which has been described in CZS. This may be worth a comment as the trajectory of the brain growth in the study animals postnatally is not known. With the findings, would the authors anticipate the development of microcephaly?

In our study, we found cortical volume relative to fetal size was decreasing as gestation progressed for several animals, which has been described in human CZS and mirrors the pathology of fetal brain disruption sequence (FBDS). We agree with the Reviewer’s comments that our experimental design with necropsy prior to term delivery may have limited our ability to detect microcephaly, and we have made changes to the text as follows:

- New text has been added to the discussion section to reflect this line of reasoning and we reference our previous work where we found that ZikV-exposed fetuses had decreases in head circumference that did not attain the threshold of <2 s.d. necessary to meet criteria for gross microcephaly (Adams Waldorf et al., Nat. Med. 2018 PMID: 29400709). The new text (lines 262-265) reads as follows:

“In the current study, ZikV-exposed fetuses did not develop gross microcephaly (>2 s.d. below age-corrected head circumference), although they did have smaller brain volumes than controls (**Table S2**), which has been reported in several other NHP models of ZikV infection in pregnancy.”

- We have added two new columns of data to Supplemental Table 2 that describe the gross anatomic measurements based on MRI findings. This includes measurements of biparietal diameter and white matter fraction of supratentorial volume at the indicated gestational ages.

13. Were there subcortical calcifications in any of the brains? Did any of the fetuses have any other features of CZS such as arthrogryposis, eye abnormalities, or were they growth restricted?

We did not find any evidence of calcifications on T2-weighted MR imaging of ZikV-exposed animals. Nor were any overt neuroanatomic abnormalities (cortical malformations, corpus callosum dysgenesis, vermian hypoplasia) noted in any of the animals. We also did not observe any eye abnormalities on MR imaging, and histopathology was not performed on the retina or optic nerves. We did not perform measurement of anatomic growth parameters or physical evaluation at necropsy

sufficient to quantify growth restriction or athrogryposis. We have added the following sentence (lines 140-142) to the Results section to clarify these observations:

“We did not observe MRI evidence of intracranial calcifications, cortical malformations, corpus callosum dysgenesis, or vermian hypoplasia in any fetus.”

14. Were any of the infected dams pregnant with a microcephalic fetus? It seems the study only includes normocephalic fetuses at CS delivery.

None of the dams infected in this study (or in related studies we have subsequently performed) have carried fetuses that developed gross microcephaly (<2 s.d. below predicted head circumference). We did not exclude any animals from this study, nor did we have dams with fetal demise after ZikV inoculation. In brief, we observed some reduction in the relative size of fetal brain of ZikV-exposed animals but it did not meet criteria for microcephaly (<2 s.d. below size predicted by gestational age). We have made changes to the text as follows:

- New text has been added to the discussion section to reflect this line of reasoning and we reference our previous work where we found that ZikV-exposed fetuses had decreases in head circumference that did not attain the threshold of 2 s.d. necessary to meet criteria for gross microcephaly (Adams Waldorf et al., Nat. Med. 2018 PMID: 29400709). The next text (lines 263-266) reads as follows:

“In the current study, ZikV-exposed fetuses did not develop gross microcephaly (>2 s.d. below age-corrected head circumference), although they did have smaller brain volumes than controls (Table S2), which has been reported in several other NHP models of ZikV infection in pregnancy.”

15. Can the authors discuss why not all fetuses had ZikV RNA detected in their brain at necropsy. Does the finding of ZikV RNA in the brain make any difference in the impact of the virus on myelin structure and OL maturation?

It is possible that the different gestational ages at ZikV inoculation may explain why virus is not detected in the fetal brain of all animals. We hypothesize that fetal infection occurred in all cases and that ZikV likely cleared the fetal brain in ZIKA3-5 due to an earlier inoculation at GD60-63, as compared to ZIKA1-2 that were inoculated later in gestation (GD119 and GD82, respectively). We acknowledge that, with only 3/6 ZikV-exposed fetuses demonstrating viral RNA, we cannot definitively conclude that the myelin phenotype we observe is due to viral infection in the fetal brain. We speculate that the consequences on myelin are secondary to direct effects of ZikV on OPC and neuron development, but there may be other potential mechanisms, including disrupted placental function, as possible explanations.

We have adjusted the text of the paper to reflect this uncertainty, and to point out several lines of evidence supporting fetal viral infection, as outlined below.

- To avoid overstating our findings, we have elected to refer to all fetuses as “ZikV-exposed”, as we confirmed maternal infection for all 6 dams.
- We have added new text in the discussion clarifying that 2/3 fetuses in which we did not detect viral RNA had MRI findings of a primary lesion in the neural progenitor niche, which we hypothesize reflects active infection in that area. The new text (lines 205-209) reads as follows:
“We were able to directly confirm ZikV infection in 3/6 fetuses, while two of three fetuses in which we did not detect ZikV RNA had MRI findings demonstrating a “primary” lesion in the posterior

periventricular region, the niche of neural progenitor cells (NPCs), arguing that they were infected with ZikV but fetal brain infection was cleared at the time of our analysis.”

- We have added text in the supplemental figure legend (Fig. S1, lines 45-46) indicating that the three animals without detectable ZikV RNA also had the longest interval after inoculation prior to necropsy. As the Reviewer points out, this may have allowed the animals to clear the virus, which appears to be the most common outcome of human fetal infection, both in fetuses that develop microcephaly and those that do not (see Oliveira et al., Int J. Gyn. Ob 2020, PMID: 31975394).
- We have clarified that an important alternate explanation that we cannot rule out is the possibility of disrupted placental physiology, which might be expected to result in fetal hypoxia or other features of placental insufficiency. We have added the following sentences in the discussion (lines 271-276) addressing this possibility:

“We have proposed a brain-intrinsic mechanism for myelin perturbation, but these experiments do not definitively rule out the possibility of extra-fetal mechanisms including maternal inflammatory cytokines or placental disruption. Although chronic placental inflammation has been associated with white matter injury in premature infants, the only placental pathology observed in ZikV-exposed animals was mild deciduitis, which was also present in some control animals.” (Adams Waldorf et al., Nat Med, 2018; Fig. S15).

- We have performed additional sub-group analyses of the transcriptomic and immunohistochemical datasets to assess whether any differences could be detected between animals with and without detectable ZikV RNA that could explain the impact of the virus on myelin structure and OL maturation. A new panel b has been added to Fig. S2 and the following sentences added to the revised manuscript:
 - i. A new sentence (below) has been added (lines 89-91) to the results section and the corresponding Fig. S2d figure legend updated on lines 62-65 of the revised Supplemental text.

“Principal component analysis did not reveal differences between animals according to detection of ZikV RNA in fetal tissue or ZikV strain inoculated, but rather tissue region was the greatest source of variation (Fig. S2d).”
 - ii. A new sentence (lines 136-138) has been added to the results section. The new sentence reads as follows:

“There were no significant differences in immunohistochemical quantification of MBP, GFAP, or Iba1 when comparing between ZikV-exposed animals based on detection of ZikV RNA in fetal tissue or ZikV strain inoculated (Fig. S1e).”

Reviewer #4 (Remarks to the Author):

In this manuscript, Tisoncik-Go et al. utilized established pigtail macaque fetal Zika virus infection model and uncovered profound disruption of fetal myelin in animals with prenatal ZikV exposure. While the overall research framework is comprehensive, further analysis of specific evidence is needed to enhance its persuasiveness.

Major concerns:

1. The authors claimed that the Zika virus exposed fetuses were non-microcephalic. However, the head circumference data and the diagnostic criteria of microcephalia in pigtail macaque were not mentioned in the text.

We did not include these data in this manuscript as they have been reported previously (Adams Waldorf et al., Nat. Med. 2018 PMID: 29400709). We acknowledge that these are important data for the evaluation of the phenotype we describe and have therefore added them to Supplemental Table 2. Table S2 has 2 new columns added that report 1) biparietal diameter for animals in which it was performed and 2) white matter fraction of supratentorial volume in animals for whom MRI data are available. There was not a significant difference in the fetal brain weight between controls and ZikV. As previously reported, there was a trend toward smaller brain volume in ZikV animals, but it did not reach the definition of gross microcephaly. We have clarified this with the following sentence (lines 71-73) in the Results section as follows:

“While there was a trend toward smaller brain volume in ZikV-exposed animals, none reached the threshold of >2 s.d. smaller than controls to be considered “microcephalic” (Table S2).”

2. The inoculation and MRI examination time points illustrated in Fig. S1a varied between maternal animals, as well as the interval between inoculation and cesarean section time, which may bring biases to the downstream analysis.

We agree that this is a possible limitation to the study, noting that we designed this study to address, in part, the time of virus challenge to identification of relevant phenotypes at necropsy. Nonetheless, it is important to note that we observed myelin perturbation as a consistent fetal brain phenotype regardless of gestational day of virus challenge of the dam. We have added new text to address this bias to the discussion section (lines 278-284). The added sentences read as follows:

“Finally, we combined analysis of animals infected with two closely-related isolates of ZikV and across a range of gestational ages from first and second trimester. We did not detect strain-related differences on sub-group analysis. Although the study may not have been sufficiently powered to detect small differences, the recapitulation of myelin decompaction in both strains and across time points argues for a conserved pathophysiologic mechanism and suggests that white matter injury may be common in human CZS.”

3. Some data seemed to be contradictory. For example, Fig S1a showed that maternal animals ZIKA 6 and Control 5 were inoculated on gestation day 118 and 134, respectively. However, in Table S1, the inoculation gestational age of ZIKA 6 was day 121 while Control 5 was day 128. The same contradiction could also be seen in Table S2, the age of ZIKA 1, CTL3 and CTL4.

We apologize for the confusion and thank the reviewer for identifying this discrepancy. This occurred due to an error in manually copying from an exported database file into Word and Illustrator format. We have corrected the discrepancies to reflect the correct data shown in Fig 1a and Table S1 of the revised manuscript.

4. Fig S1e showed that, RNA of Zika virus was not found in the brains of fetuses ZIKA 3, 4, 5, and authors did not put forward any other data to prove the fetuses were infected by Zika virus. Whether fetus modeling succeeded remained to be prove.

The Reviewer is highlighting the discrepancy between ZikV RNA detection in 3/6 fetuses while myelin perturbation was detected (by at least one assay) in all ZikV-exposed fetuses. This is an important

point and we have made significant changes in the manuscript to reflect our consideration. We feel that infant infection was likely in all animals because of corroborating evidence involving the following:

- MRI evidence of a “primary” lesion consistent with ZikV infection of the neural progenitor niche in 2/3 animals without ZikV RNA detected
- A longer duration between inoculation and necropsy in the three animals without ZikV RNA detected, perhaps allowing the animals to clear the infection.
- The finding among human neonates with CZS that detection of viral RNA in tissues occurs in less than half of cases, particularly in those with milder symptoms (Oliveira et al., *Int J. Gyn. Ob* 2020, PMID: 31975394).

We acknowledge that we cannot definitively establish primary ZikV infection in 3/6 fetuses, though we confirmed maternal infection for all 6 dams. We hypothesize that fetal infection occurred in all cases and ZIKA3, 4 and 5 likely cleared the infection due to an earlier maternal inoculation at GD60-63 and the longer interval until necropsy. As such, we cannot definitively conclude that the myelin phenotype we observe is due to primary viral infection in the fetal brain and we acknowledge that other potential mechanisms, including disrupted placental function, are possible explanations. We have adjusted the text of the paper to reflect this uncertainty, and to point out several lines of evidence supporting fetal viral infection, as outlined below.

- First, we have added text in the discussion (lines 205-209) clarifying that 2/3 fetuses in which we did not detect viral RNA had MRI findings of a primary lesion in the neural progenitor niche, which we hypothesize reflects active infection in that area.

“We were able to directly confirm ZikV infection in 3/6 fetuses, while two of three fetuses in which we did not detect ZikV RNA had MRI findings demonstrating a “primary” lesion in the posterior periventricular region, the niche of neural progenitor cells (NPCs), arguing that they were infected with ZikV but fetal brain infection was cleared at the time of our analysis.”

- Second, we have added text in the supplemental figure legend (Fig. S1, lines 45-46) indicating that the three animals without detectable ZikV RNA also had the longest interval after inoculation prior to necropsy. As the reviewer points out, this may have allowed the animals to clear the virus, which appears to be the most common outcome of human fetal infection, both in fetuses that develop microcephaly and those that do not (see Oliveira et al., *Int J. Gyn. Ob* 2020, PMID: 31975394).
- Third, we have clarified that an important alternate explanation that we cannot rule out is the possibility of disrupted placental physiology, which might be expected to result in fetal hypoxia or other features of placental insufficiency. We have added the following sentences in the discussion (lines 271-276) addressing this possibility:

“We have proposed a brain-intrinsic mechanism for myelin perturbation, but these experiments do not definitively rule out the possibility of extra-fetal mechanisms including maternal inflammatory cytokines or placental disruption. Although chronic placental inflammation has been associated with white matter injury in premature infants, the only placental pathology observed in ZikV-exposed animals was mild deciduitis, which was also present in some control animals.” (Adams Waldorf et al., *Nat Med*, 2018; Fig. S15).

5. The authors claimed in the abstract that Zika virus exposed animals showed perturbation or remodeling of previously formed myelin. However, the conducted experiments demonstrated a substantial downregulation in gene expression related to crucial components of oligodendrocyte maturation and

showcased a disruption in myelination. Notably, there is an absence of evidence supporting the claim of remodeling of pre-formed myelin. To address this gap, the inclusion of new time points in the experimental design is recommended. This additional temporal dimension will enable a clear differentiation between the remodeling of previously formed myelin and the disruption of myelin formation.

We agree with the Reviewer's astute comment that we cannot definitely ascribe the myelin decompaction phenotype to a loss of previously-formed myelin. Our reasoning for this is based on two features of the disrupted myelin we observed in ZikV-exposed fetuses. First, that there were areas of intact myelin on most axons that had similar features (number of wraps, wrap thickness) to compact myelin in control animals. Second, that the gross appearance of myelin decompaction most closely resembled phenotypes seen in axon injury models such as optic nerve crush, and less like models of altered oligodendrocyte development such as mice with knockout of PDGF signaling (Fruttiger et al., *Devel.* 1999, PMID: 9876175).

While we acknowledge that the addition of samples from earlier time points might help answer this question, unfortunately we do not have animals in this study with a shorter latency to inoculation or an earlier age at necropsy. However, we point out that we were able to perform limited time course analysis on the animals used in electron microscopy as reflected in Figure S7. Of note, these animals demonstrate predicted age-related changes in myelin including increasing fraction of axons myelinated, increasing number of wraps, and decreasing wrap thickness. Together, these findings suggest that many features of myelin are undergoing appropriate developmental progression. It remains possible that decompaction is due to abnormal development of myelin. We have made the following changes to the manuscript to reflect this reasoning:

- First, to emphasize the temporal distribution of sampling we updated this sentence (lines 166-167) to read as follows:
“Intact regions of myelin had apparently normal ultrastructural properties, including number of wraps and wrap thickness as expected based on gestational age”.
- Second, we tempered our conclusions regarding the mechanism of myelin decompaction by removing the mechanistic implications from the sentence (lines 8-10) in the abstract. The revised sentence now reads as follows:
“At the ultrastructural level, the myelin sheath in ZikV-exposed animals showed multi-focal decompaction, occurring concomitant with dysregulation of oligodendrocyte gene expression and maturation.”

6. Further endeavors in transcriptomic data analysis could be undertaken to elucidate the relationship between neuronal maturation and synaptic formation.

We appreciate the insight of the Reviewer in highlighting the possible links between neuronal maturation, synapse formation, and myelin perturbation. We performed additional analysis of the spatial transcriptomic data to provide a deeper mechanistic understanding of ZikV-related changes to neurodevelopment. This new network analysis identified three pathways (Synaptic signaling, Neuron projection guidance, and Neuron differentiation), bolded in Fig. S2b. The gene relationships among these three pathways are displayed in a network that is now included in a new panel in Fig. S2c. Our analysis shows that genes underlying neuron differentiation are generally downregulated in deep grey matter, while genes that are enriched in immature neurons (e.g., *SATB2*, *SOX11*, and *DCX*) are upregulated in deep grey matter. This suggests that neurons in ZikV-exposed fetuses are less mature than controls. In support of this, the network representing synaptic signaling shows a negative enrichment score and several genes

for key components of presynaptic function (e.g., *CPLX1*, *SLC17A7*, *CACNB4*). Together, these findings argue that neurons in ZikV-exposed animals may have an immature phenotype with increased axon outgrowth and incomplete synapse formation. We hypothesize that this immaturity contributes to a loss of appropriate signaling between neurons and oligodendrocytes leading to decompaction of myelin.

- We have revised the text in the results (lines 88-92) emphasizing key genes from neuron differentiation that are upregulated in deep grey matter and citing the new network in Fig. S2c. The revised sentence reads as follows:

“In contrast, the grey matter of ZikV-exposed fetal brains showed increased expression (upregulation) of genes underlying axon growth (*NCAM1*, *TUBB*, *GAP43*; **Fig. 1e**), and neuronal immaturity (*SOX11*, *DCX*, *SATB2*; **Fig. S2c**) compared to control.”

- We have outlined this hypothesis in the discussion (lines 248-253). The new sentences read as follows:

“Oligodendrocyte maturation and myelin synthesis are closely coupled to neuronal maturation and function in a bidirectional manner. Therefore, we propose that the disruption of myelin may be related to a loss of trophic or maturation signals derived from local neurons or even astrocytes (**Fig. S8**). Indeed, our spatial transcriptional data from deep grey matter shows a decrease in expression of genes for synaptic function (*CPLX1*, *SLC17A*) and an increase in genes associated with immature neurons (*SOX11*, *DCX*, *SATB2*).”

Minor concerns:

7. On the line 70 of the text, authors mentioned that transient viremia was found in 6/7 dams while there were only 6 maternal animals in total and only 5 found virus RNA in plasma. Also, the figure reference should be Fig. S1e rather than Fig. S1d.

We appreciate the careful reading by the Reviewer and have corrected the sentence (lines 69-71) that now reads as follows:

“Within the ZikV challenge cohort, transient viremia was demonstrated across 5/6 ZikV-challenged dams at 2 days post-infection (DPI), with ZikV RNA detected in fetal brain at necropsy of 3/6 ZikV cohort animals (**Fig. S1e**).”

8. Fig S1c showed an horizontal brain section, the figure legend annotated it as a coronal plane.

We appreciate the point and the careful reading by the Reviewer and we have corrected the Fig. S1 figure legend text (line 31) that now reads as follows:

“**c**) Fetal cerebrum bulk tissue dissection scheme overlaid onto an MRI of a control animal at 156 gestational days (GD) in the horizontal plane.”

9. On the line 76 of the text, words ‘grey matter’ should be ‘deep grey matter’, according to the abbreviation and Fig 1a.

We appreciate the careful reading by the reviewer and we have corrected this sentence (line 77) that now reads as follows:

“We chose ROIs representing functionally distinct compartments as follows: deep grey matter (DGM, containing cortical Layer V pyramidal neuron cell bodies), superficial white matter (SWM, containing proximal axons in cortical Layer VI), and deep white matter (DWM, containing myelinated axons of projecting neurons deep to the cortex) (**Fig.1a**).”

REVIEWERS' COMMENTS

Reviewer #1 (Remarks to the Author):

This is a substantially revised manuscript that was highly and positively responsive to the reviewers comments. Nearly all comments were addressed to this reviewers satisfaction and the findings will be of wide interest in the field of vertically transferred viruses. This reviewer has no further comments.

Reviewer #1 (Remarks on code availability):

yes, the paper is reproducible; I was able to install and run the code. Which on a Mac computer was a surprise.

Reviewer #3 (Remarks to the Author):

Thank you for addressing my comments in your revised manuscript.

Reviewer #4 (Remarks to the Author):

The revised manuscript has addressed the majority of concerns, and I have no further comments to add.